# Closed-loop Long-horizon Robotic Planning via Equilibrium Sequence Modeling

## Abstract

In the endeavor to make autonomous robots take actions, task planning is a major challenge that requires translating high-level task descriptions into long-horizon action sequences. Despite recent advances in language model agents, they remain prone to planning errors and limited in their ability to plan ahead. To address these limitations in robotic planning, we advocate a self-refining scheme that iteratively refines a draft plan until an equilibrium is reached. Remarkably, this process can be optimized end-to-end from an analytical perspective without the need to curate additional verifiers or reward models, allowing us to train self-refining planners in a simple supervised learning fashion. Meanwhile, a nested equilibrium sequence modeling procedure is devised for efficient closed-loop planning that incorporates useful feedback from the environment (or an internal world model). Our method is evaluated on the VirtualHome-Env benchmark, showing advanced performance with better scaling for inference computation. Code is available at `https://github.com/anonymous-iclr-2025/equilibrium-planner`.

## 1 Introduction

Recent advances in large language models (LLMs) have spurred tremendous progress in robotic planning (Huang et al., 2022; Li et al., 2022; Singh et al., 2023; Driess et al., 2023; Ahn et al., 2023; Huang et al., 2023; Zhao et al., 2023; Hu et al., 2024). Based on their extensive world knowledge, LLM agents seem close to autonomously performing robotic tasks, such as in household scenarios. However, growing evidence shows that existing LLM agents struggle with task planning (Kaelbling & Lozano-Pérez, 2011) that decomposes a high-level task into mid-level actions. While this problem requires long-horizon planning as well as consideration of environmental feedback, LLMs are often limited by: (1) *unidirectional dependency*: due to autoregressive generation, previous tokens cannot attend to future tokens, resulting in limited ability to plan ahead (Wu et al., 2024a); (2) *lack of error correction* for existing outputs, unless with a heavy system 2; (3) *fixed forward process* hindering the allocation of more inference computation to further improve planning performance. These inherent limitations of LLMs lead to inefficiency in the closed-loop long-horizon robotic planning.

To address above challenges of LLM planners in closed-loop long-horizon planning, we advocate the approach of self-refinement (Welleck et al., 2023; Shinn et al., 2023; Kim et al., 2023; Madaan et al., 2023) that iteratively improves a previously generated plan. The reasons behind are threefold: (1) *bidirectional dependency*: since the output is conditioned on a previous draft plan, it can attend to all tokens in the plan (from an old version), thus improving its ability to plan ahead; (2) *internal error correction* which allows implicit self-correction in a forward pass without an explicit, heavy system 2; (3) *dynamic computation allocation* by iterating through a self-refinement process until convergence. However, such a self-refining strategy imposes significant training difficulties because it requires backpropagation through infinite self-refining steps (Werbos, 1990). This may be seen as an extreme case that reflects some of the general challenges in teaching LLMs to plan and reason. Existing solutions include curating process supervision (Uesato et al., 2022; Lightman et al., 2024) or applying reinforcement learning (Zelikman et al., 2024; OpenAI, 2024; Kumar et al., 2024), but they are considerably more complex than supervised training and difficult to implement.

This work proposes a simple learning framework for planning via self-refinement. Specifically, we formulate the self-refining process as a fixed-point problem that recursively refines the plan until the equilibrium point, as illustrated in Fig. 1. While the forward process of this fixed-point problem

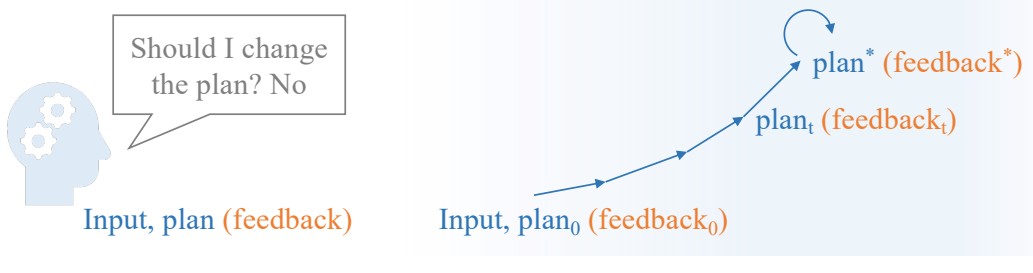

(a) Equilibrium in planning        (b) Iterative planning until an equilibrium is reached

Figure 1: Illustration of the equilibrium point in planning. We view planning as a self-refinement process in which the ideal plan emerges as an equilibrium point, remaining unchanged by any refinement attempts even with newer information (*e.g.* feedback from the environment or a world model). This enables us to tackle robotic planning from an optimization perspective around its equilibrium.

could be solved efficiently using root-finding methods, more interestingly, its backpropagation can be *skipped* since its gradient is explicated by the implicit function theorem (Krantz & Parks, 2002) as in deep equilibrium models (Bai et al., 2019; Geng & Kolter, 2023). It is noted that the derived gradient term may be further simplified through a Jacobian-free approximation (Fung et al., 2022) to facilitate training. These analytical techniques allow end-to-end supervised training of the LLM planner to accomplish self-refinement without the need for additional verifiers or reward models in reinforcement learning-based counterparts, greatly enhancing simplicity and practicality. And after training, our equilibrium model-based planner is capable of dynamically allocating more inference computation in the equilibrium solving process to achieve better planning performance.

Another important cue for self-refinement in robotic tasks is closed-loop feedback from the environment. To efficiently incorporate environmental feedback, we devise a nested equilibrium sequence modeling procedure consisting of inner and outer loops, where the inner loop iteratively refines a plan using previous feedback, while the outer loop updates the feedback by interacting with the environment. This enables closed-loop planning from even a few environmental interactions. Moreover, the nested equilibrium solving process is accelerated by reusing the previously derived equilibrium. We further implement the above design within an LLM agent framework, seamlessly integrating the equilibrium model-based planner, an experience memory buffer containing past plans and feedback, and a world model to estimate feedback in the absence of environmental interactions, thus allowing the planning system to operate effectively in closed-loop long-horizon robot task planning scenarios. The core contributions of our work are summarized as follows:

- We present equilibrium sequence modeling, a simple training approach for self-refining LLMs based on equilibrium models, allowing for end-to-end learning in a supervised manner.
- A nested equilibrium solving process is proposed to efficiently incorporate closed-loop feedback into the equilibrium sequence modeling, reusing previous equilibrium solutions to alleviate inference computation. It is further implemented with a world model to improve practicality.
- Our method is evaluated on the VirtualHome-Env benchmark (Puig et al., 2018; Liao et al., 2019), demonstrating its advantageous performance with better scaling w.r.t. inference computation.

## 2 RELATED WORK

**LLMs for Planning.** Scaling up inference computation to improve LLMs' performance on planning and reasoning tasks has received increasing attention (Brown et al., 2024; Snell et al., 2024; Wu et al., 2024b; OpenAI, 2024). Existing techniques involving chain-of-thought (Wei et al., 2022; Zelikman et al., 2022; 2024), repeated sampling (Wang et al., 2023; Brown et al., 2024) and tree search (Yao et al., 2023a; Zhao et al., 2023) showed preliminary results with handcraft system 2. Alternatively, a method called self-refinement (Welleck et al., 2023; Shinn et al., 2023; Kim et al., 2023; Madaan et al., 2023) suggests recursively refining the existing LLM output in an autonomous manner, but it relies heavily on prompting or intricate training procedures. To fully exploit its potential, we propose an end-to-end optimization method for self-refinement through deep equilibrium models.

**Deep Equilibrium Models** (Bai et al., 2019) are infinite-depth neural networks specified by fixed-point problems $x^* = f_\theta(x^*)$, where $f_\theta$ is an equilibrium layer. While their inference can take infinite steps by the fixed-point iteration, their gradients are estimated using implicit differentiation (Krantz & Parks, 2002) without backpropagating through all layers, thus enabling memory-efficient training. They have been extensively applied to tasks such as visual understanding (Bai et al., 2020; 2022) and image generation (Pokle et al., 2022; Geng et al., 2023; Bai & Melas-Kyriazi, 2024). In this paper, we apply the fixed-point formulation of deep equilibrium models to the self-refinement process in LLM planners, allowing for simple supervised training of LLMs to refine themselves. More detailed introduction to deep equilibrium models is presented in Appendix A.

## 3 METHOD

We study the problem of robot task planning which aims to decompose a high-level task description into long-horizon mid-level action sequences (Kaelbling & Lozano-Pérez, 2011). In the following, we first discuss the limitations of LLM planners in self-refinement (Section 3.1) and address them with a novel equilibrium sequence modeling scheme (Section 3.2), which is compatible with various feedback from the designs in Section 3.3. Lastly, practical implementations are given in Section 3.4.

### 3.1 PRELIMINARIES ON SELF-REFINEMENT

The prevailing LLMs are intrinsically limited in planning, as their unidirectional dependency results in limited capability to plan ahead (Wu et al., 2024a), and the lack of error correction hinders closed-loop planning. These reasons call for alternative mechanisms to address robot task planning.

Recently, Welleck et al. (2023); Shinn et al. (2023); Kim et al. (2023); Madaan et al. (2023) proposed self-refinement, which uses an LLM $f_\theta$ to iteratively refine the previous LLM output. This strategy naturally addresses the above limitations, since it introduces bidirectional token dependency and a dynamic error correction mechanism. Formally, let $x_t$ denote a draft plan and $c_t$ denote context (*e.g.* environmental feedback), then planning may be viewed as a self-refinement process as follows:

$$x_{t+1} = f_\theta(x_t, c_t). \tag{1}$$

However, self-refinement via prompting (Shinn et al., 2023; Kim et al., 2023; Madaan et al., 2023) has been found to be very limited by Huang et al. (2024). Alternative training-based methods require careful curation of training sequences (Welleck et al., 2023; Havrilla et al., 2024) or reinforcement learning (Qu et al., 2024; Kumar et al., 2024) and are therefore difficult to train. Overall, they remain deficient for robotic planning compared to system 2-based alternatives, as shown in Hu et al. (2024).

### 3.2 SELF-REFINEMENT AS AN EQUILIBRIUM MODEL

To address the training inefficiency of self-refinement approaches, this section proposes equilibrium sequence modeling, a simple supervised training scheme for teaching LLM planners to self-refine through the lens of deep equilibrium models (Bai et al., 2019; Geng & Kolter, 2023).

Let us first consider a simplified scenario of self-refinement without environmental feedback, namely that the context $c_t$ is fixed, *e.g.* to a predefined system message $c$. Then, the self-refinement process in Eq. (1) reduces to a fixed-point problem concerning only the plan $x_t$. Denote the initial plan by $x_0 = \varnothing$ and the equilibrium plan, *i.e.* the endpoint, by $x^*$, then its trajectory can be expressed as:

$$(x_0, c) \to \ldots \to (x_t, c) \to \ldots \to (x^*, c). \tag{2}$$

Although its forward process is tractable with existing root-solving techniques, such as the classic fixed-point iteration or alternative numerical methods (Broyden, 1965; Anderson, 1965), its training requires recurrent backpropagation through multiple self-refining steps (Werbos, 1990). This results in an extremely inefficient and unstable computational process where end-to-end training fails.

Instead, we approach it directly from an analytical perspective. Assuming access only to an outcome supervision $L(\cdot, y)$ on the plan, *e.g.* its distance to the ground truth plan $y$, then self-refinement is formulated as an optimization problem minimizing the loss function of the equilibrium plan:

$$\min_\theta \quad L(x^*, y)$$
$$\text{s.t.} \quad x^* = f_\theta(x^*, c). \tag{3}$$

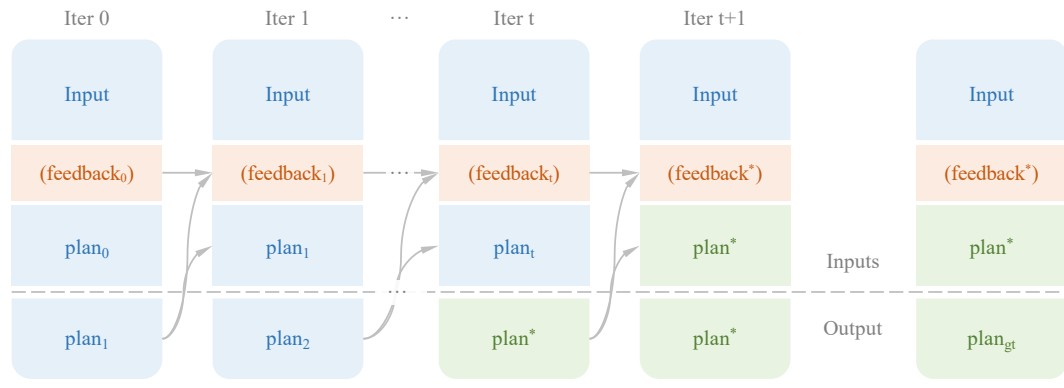

(a) Inference: solving equilibrium via fixed-point iteration    (b) Training

Figure 2: Illustration of equilibrium sequence modeling. Prior to training, our model first undergoes iterative inference to reach an equilibrium plan. It is then pushed away from the equilibrium towards ground truth by a supervised training loss. This process enables self-refinement of the model.

Interestingly, the above optimization problem can be solved without backpropagating over the entire inference process. As the following theorem indicates, we can directly differentiate through its fixed point regardless of the solution path, with only a constant computational and memory cost.

**Theorem 1** *(Implicit Function Theorem (Bai et al., 2019; Krantz & Parks, 2002)) Assuming that* $\left(I - \frac{\partial f_\theta}{\partial x^*}\right)$ *is invertible, then the loss gradient of Eq.* (3) *w.r.t.* $\theta$ *is given by:*

$$\frac{\partial L}{\partial \theta} = \frac{\partial L}{\partial x^*} \left(I - \frac{\partial f_\theta}{\partial x^*}\right)^{-1} \frac{\partial f_\theta}{\partial \theta}. \tag{4}$$

Its proof is given in Appendix A.4. It is noteworthy that the inverse Jacobian term $A = (I - \frac{\partial f_\theta}{\partial x^*})^{-1}$ within the above gradient is difficult to compute exactly, for which existing work often approximates through the damped fixed-point unrolling or the Neumann series (Geng et al., 2021b). For computational efficiency, we drop the inverse Jacobian term using $A \approx I$ as in Fung et al. (2022); Geng et al. (2021a); Choe et al. (2023), the latter having been validated on Transformer-based LLMs:

$$\frac{\partial L}{\partial \theta} = \frac{\partial L}{\partial x^*} \frac{\partial f_\theta}{\partial \theta}. \tag{5}$$

**Equilibrium Sequence Modeling.** Based on the simplified gradient estimation, we reformulate its training into a supervised learning problem. According to the chain rule, the derived gradient is exactly the gradient of the following optimization problem associated with the equilibrium $x^*$:

$$\min_\theta \quad L(f_\theta(x^*, c), y). \tag{6}$$

This new formula represents a new equilibrium sequence modeling scheme that can be implemented in two stages as in Fig. 2. In the first stage, we iterate over the fixed-point problem within Eq. (3) to solve the equilibrium plan $x^*$. Then, it is paired with the ground truth plan $y$ as a training sequence, which is used as in the standard supervised finetuning pipeline to teach the LLM to self-refine.

It features two intuitive advantages: (1) instead of directly regressing the ground truth, it only adjusts the equilibrium point, which reduces overfitting compared to the vanilla supervised finetuning; (2) it teaches self-refinement by a simple supervised loss, without requiring additional value functions or reward models (Welleck et al., 2023; Havrilla et al., 2024; Qu et al., 2024; Kumar et al., 2024).

### 3.3 EQUILIBRIUM MODELS WITH FEEDBACK

This section extends the derived equilibrium sequence modeling to a more practical scenario where the environment may provide some closed-loop feedback, *e.g.* failure details, during plan execution. Such auxiliary information would be an effective cue for planners to further refine their plan.

---

**Algorithm 1** Inference of Equilibrium Planner

---

**Require:** planner $f_\theta$, environment or world model Env, number of iterations $N$.
  Initialize a start point $x_0$ and feedback $c_0$.
  **for** $i \leftarrow 0$ to $N$ or converged **do**
    Solve inner equilibrium loop to obtain $x_t^*$.            ▷ Eq. (8)
    Update next plan $x_{t+1}$ and feedback $c_{t+1}$ with Env.    ▷ Eqs. (9) and (10)
**Ensure:** generated plan $x^*$.

---

To take into account environmental feedback, we consider an adaptive context $c_t$ that is influenced by the plan $x_t$ rather than fixed. Then, the previously concerned equilibrium solving process of Eq. (2) should be revised as an iterative process that couples plan and feedback, starting from $x_0 = c_0 = \varnothing$:

$$(x_0, c_0) \rightarrow \ldots \rightarrow (x_t, c_t) \rightarrow \ldots \rightarrow (x^*, c^*). \tag{7}$$

After the modification, the existing derivations only hold when we neglect the derivatives related to $c^*$. Fortunately, this is a natural choice due to the non-differentiability of most feedbacks. Therefore, the equilibrium planner can be trained in a similar supervised way as in Eq. (6) and Fig. 2, and after training it would be able to self-refine based on the latest feedback just by forward passes. However, iteratively interacting with the environment to obtain feedback is costly and may not be recoverable. In response, we devise a nested equilibrium solving scheme for more efficient closed-loop planning.

**Nested Equilibrium Solving.** Inspired by the introspection process in human daily life, we propose to divide equilibrium solving into a nested loop process. The inner loop *introspects* on the existing plan and feedback and takes no action, while the outer loop interacts with the environment to update the feedback. Formally, each inner loop is an equilibrium solving process with fixed feedback $c_t$:

$$\begin{cases} x_t^1 = f_\theta(x_t^0, c_t) \\ \ldots \\ x_t^* = f_\theta(x_t^*, c_t). \end{cases} \tag{8}$$

Thanks to this inner-loop introspection mechanism, our equilibrium model can be more efficient in closed-loop planning, achieving superior performance with fewer environmental interactions.

**Reusing Equilibrium Solution.** Another efficiency bottleneck is the equilibrium solving. Considering that its speed depends largely on the initial plan, it is unnecessary to restart from $\varnothing$ every time. Therefore, we accelerate equilibrium solving by reusing the previously derived equilibrium plan as the starting point of the next iteration, similar to Bai et al. (2022); Bai & Melas-Kyriazi (2024):

$$x_{t+1} = x_t^*. \tag{9}$$

which corresponds to the starting point $x_{t+1}^0$ of the inner loop. Similarly, history feedback could be reused across different inner loops. This is achieved by initializing the context of the next inner loop by concatenating the previous feedback with the latest feedback (paired with its original plan):

$$c_{t+1} = (x_{t+1}, \text{Env}(x_{t+1})) \parallel c_t, \tag{10}$$

where $\parallel$ denotes concatenation. The whole nested inference procedure is described in Algorithm 1.

## 3.4 PRACTICAL IMPLEMENTATION

This section discusses the implementation of the proposed equilibrium planner. To enable effective training while interacting with the environment, the following two modules are carefully devised to complement the planner: an experience memory that caches all equilibrium plans and their feedback during equilibrium solving, and a world model to estimate the closed-loop feedback in the absence of environmental interactions. The planning framework is illustrated in Fig. 3.

**Equilibrium Experience Memory.** During the training process, our equilibrium model interacts with the environment only when the inner loop reaches the equilibrium point. This results in a small number of equilibrium points, which may not be sufficient for supervised training. To improve our training efficiency and stability, we opt to cache all previously obtained equilibrium points, along with their environmental feedback, in an experience memory. Thereafter, these equilibrium points

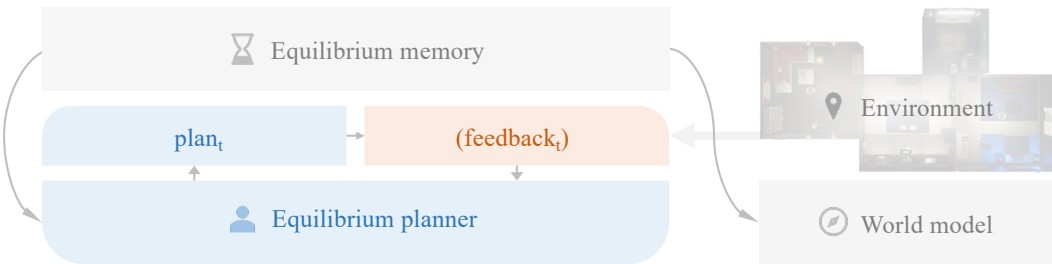

Figure 3: Illustration of our proposed framework. It incorporates (1) a memory containing all the equilibrium experience during inference, (2) a self-refining planner trained on equilibrium plans with the ground truth, and (3) a world model trained on the memory to simulate environmental feedback.

can be sampled repeatedly for versatile training purposes. For example, for the planner, we randomly sample a batch of equilibrium points at each training epoch, which are paired with the ground truth for supervised training. In particular, the most recent equilibrium points are sampled more frequently to reduce distribution shift. Next, we describe another crucial component of our framework.

**Internal Feedback from World Model.** Due to inefficiency of interacting with the environment, we construct a world model (Ha & Schmidhuber, 2018) to provide the necessary feedback in closed-loop planning. Our world model takes the environmental context, task instruction and current plan as inputs and predicts some basic types of feedback. This definition is slightly simpler than the commonly used world model, which requires simulation of the environmental states, and therefore may be easier to train. Concretely, we implement the world model with an LLM and finetune it on the planner's equilibrium feedback over all iterations for better generalizability. And during inference, we alternate between using external and internal feedback in closed-loop robotic planning.

## 4 EXPERIMENTS

### 4.1 EXPERIMENTAL SETTINGS

**Benchmark.** The VirtualHome-Env benhmark (Puig et al., 2018; Liao et al., 2019) is adopted during the experiments. It contains 1360 long-horizon tasks with ground truth action sequence annotations (average length 10.8) and provides updated scene graphs after each action, allowing simulation of closed-loop feedback. To analyze the generalizability, we divided the dataset into a training set and three test subsets, including the novel scene set, the novel task set, and the novel scene and task set. More statistics and examples about VirtualHome-Env can be found in Appendix B.1.

**Metrics.** We use executability (Exec.), success rate (SR), goal conditions recall (GCR) following Hu et al. (2024). Exec. evaluates whether the plan can be executed in given the environment, SR refers to whether the goal is accomplished, and GCR measures the proportion of goal conditions achieved. To examine the closed-loop planning capabilities, we study two test setups, without error correction or with up to 10 corrections, the latter allowing for self-correction based on environmental feedback. We also evaluate the computational efficiency by measuring TFLOPS at inference time.

**Baselines.** Our method is compared with Tree-Planner (Hu et al., 2024), SELF-REFINE (Madaan et al., 2023), and a supervised finetuned planner. They are all reproduced using Llama 3 8B (Dubey et al., 2024). In addition, we consider several baseline methods that call the GPT-3.5 API, including ProgPrompt (Singh et al., 2023), Zero-shot Planner (Huang et al., 2022) and two self-refining planners, Local Replan (Raman et al., 2022; Guo et al., 2023) and Global Replan (Shinn et al., 2023). Their results are presented for reference only. See Appendix B.2 for more details.

**Implementation Details.** Our implementation is consistent with the baseline methods by finetuning from Llama 3 8B (Dubey et al., 2024) on the VirtualHome-Env training set (paired with the equilibrium points). The number of finetuning epochs is set to 6, and the learning rate is set to 0.0002. The world model is finetuned on all planner interactions for 5 epochs using the same learning rate. A greedy LLM sampling strategy is used in later refinement steps until convergence. Moreover, we implement the KV cache to speed up inference. Further details are provided in Appendix B.3.

Table 1: Performance on VirtualHome-Env without correction. Our planner achieves state-of-the-art performance in most evaluations. Note that the Exec. metrics are marked in gray because they are already high and can easily exceed 99% with simple automated rules (by truncating illegal output).

| | Novel Scene and Task | | | Novel Scene | | | Novel Task | | |
|---|---|---|---|---|---|---|---|---|---|
| | Exec. | SR | GCR | Exec. | SR | GCR | Exec. | SR | GCR |
| *GPT-3.5 API:* | | | | | | | | | |
| Zero-shot Planner | 16.49 | 1.07 | 1.52 | - | - | - | - | - | - |
| ProgPrompt | 35.04 | 12.54 | 19.99 | - | - | - | - | - | - |
| Iterative-Planner | 44.54 | 27.04 | 33.25 | - | - | - | - | - | - |
| Tree-Planner$_{N=25}$ | 55.74 | 28.33 | 39.96 | - | - | - | - | - | - |
| Tree-Planner$_{N=50}$ | 49.01 | 28.14 | 35.84 | - | - | - | - | - | - |
| *Finetuned Llama 3 8B:* | | | | | | | | | |
| Supervised | 93.55 | 24.19 | 32.55 | 96.84 | 41.05 | 49.81 | 95.94 | 26.07 | 35.53 |
| Tree-Planner$_{N=25}$ | 95.16 | 38.71 | 63.18 | 96.08 | 51.58 | 69.45 | 95.50 | 40.38 | **63.75** |
| Tree-Planner$_{N=50}$ | 94.94 | 38.71 | 63.50 | 96.06 | 51.58 | 69.54 | 95.40 | 39.74 | 63.29 |
| Ours | 90.32 | **40.32** | **65.40** | 95.79 | **65.26** | **79.47** | 93.38 | **41.88** | 62.76 |

Table 2: Performance on VirtualHome-Env with up to 10 corrections. Our planner consistently leads in SR and GCR performance. The Exec. metrics are shown in gray for the same reasons in Table 1.

| | Novel Scene and Task | | | Novel Scene | | | Novel Task | | |
|---|---|---|---|---|---|---|---|---|---|
| | Exec. | SR | GCR | Exec. | SR | GCR | Exec. | SR | GCR |
| *GPT-3.5 API:* | | | | | | | | | |
| Local Replan | 79.66 | 37.46 | 51.90 | - | - | - | - | - | - |
| Global Replan | 82.09 | 37.93 | 52.46 | - | - | - | - | - | - |
| Tree-Planner$_{N=25}$ | 89.13 | 35.30 | 56.65 | - | - | - | - | - | - |
| Tree-Planner$_{N=50}$ | 88.26 | 41.58 | 59.55 | - | - | - | - | - | - |
| *Finetuned Llama 3 8B:* | | | | | | | | | |
| SELF-REFINE | 96.77 | 43.55 | 65.18 | 92.63 | 54.74 | 70.24 | 94.44 | 39.96 | 62.37 |
| Tree-Planner$_{N=25}$ | 95.16 | 41.94 | 56.49 | 96.08 | 55.79 | 68.82 | 95.50 | 42.09 | 57.83 |
| Tree-Planner$_{N=50}$ | 94.94 | 43.55 | 58.91 | 96.06 | 58.95 | 70.00 | 95.40 | 43.38 | 59.79 |
| Ours | 91.94 | **56.45** | **76.63** | 97.89 | **77.89** | **87.07** | 92.31 | **54.91** | **74.18** |

## 4.2 Main Results

The experimental results in the two planning setups, without correction or with up to 10 corrections, are summarized in Tables 1 and 2. Overall, our method achieves the leading performance on the majority of metrics. Specifically, the experimental results show that: (1) Even without error correction, our self-refining process still brings a significant improvement of 14% on SR in the novel scene subset, with other metrics superior or comparable to the previous leading method. (2) By incorporating environmental feedback, our approach improves all metrics by more than 11% and up to 19%, showing clear advantages. (3) Similar to the existing finetuning-based methods, our generated plans exhibit a high executability of over 90%, which can be improved to 99% by simply truncating illegal overlength outputs. These results clearly confirm the advantages of our approach.

Figure 4 further compares our self-correction process with the baseline methods. As can be seen, our method is better at incorporating environmental feedback to improve the plan, while the baselines fail by simply repeating the previous plan, or by making only local adjustments that are insufficient. This exemplifies the effectiveness of equilibrium sequence modeling for self-correction.

## 4.3 Ablation Study

**Effectiveness of equilibrium sequence modeling** is quantitatively verified in Table 2. Compared to alternative methods, our approach shows more than 11% improvement over the prompting-based

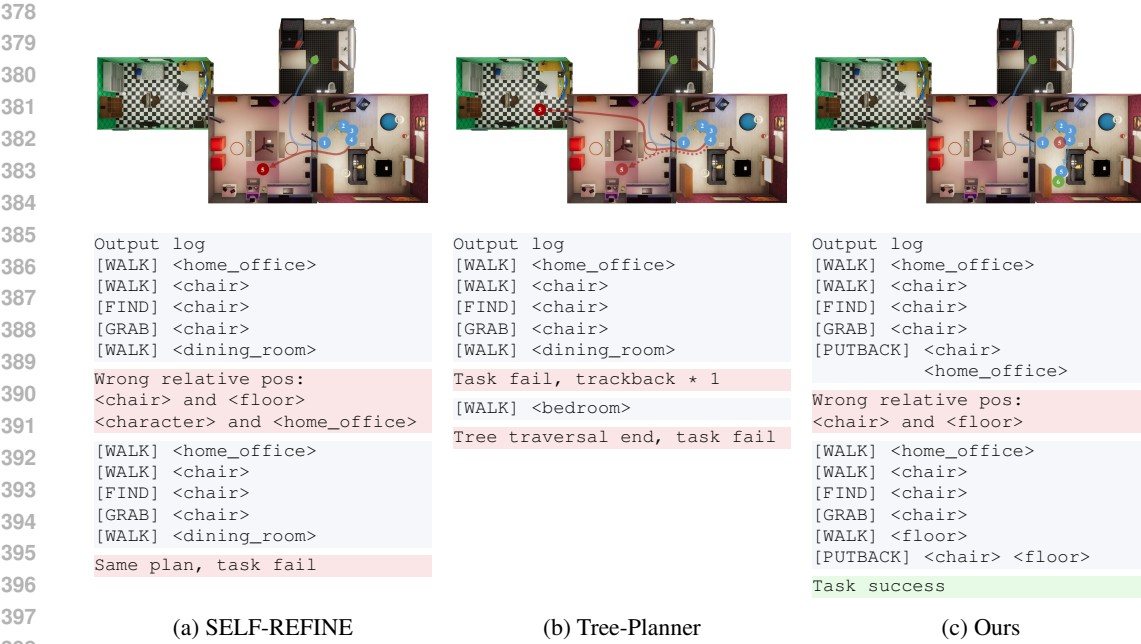

(a) SELF-REFINE         (b) Tree-Planner         (c) Ours

Figure 4: Visualization of our self-correction process compared to the baselines SELF-REFINE and Tree-Planner. The task instruction is "Take a comfortable chair and place it in the entrance hall".

Table 3: Effectiveness of different types of feedback. They are measured under the constraint of up to 10 rounds of internal or external feedback. The last two rows are reported in the main tables.

| World model | Env. | Novel Scene and Task | | | Novel Scene | | | Novel Task | | |
| --- | --- | --- | --- | --- | --- | --- | --- | --- | --- | --- |
| | | Exec. | SR | GCR | Exec. | SR | GCR | Exec. | SR | GCR |
| | | 88.71 | 33.87 | 59.98 | 96.79 | 49.47 | 66.60 | 93.80 | 34.62 | 59.06 |
| | ✓ | 83.87 | 51.61 | 75.13 | 96.84 | 75.79 | 85.79 | 92.31 | **56.62** | **75.53** |
| ✓ | | 90.32 | 40.32 | 65.40 | 95.79 | 65.26 | 79.47 | 93.38 | 41.88 | 62.76 |
| ✓ | ✓ | 91.94 | **56.45** | **76.63** | 97.89 | **77.89** | **87.07** | 92.31 | 54.91 | 74.18 |

self-refinement method SELF-REFINE (Madaan et al., 2023) and more than 12% improvement over the system 2-based Tree-Planner (Hu et al., 2024). Meanwhile, we maintain a simplistic supervised finetuning fashion similar to the compared methods, without intricate reinforcement learning. This validates the effectiveness of equilibrium sequence modeling in our robot task planning problem.

**Effectiveness of various feedback.** As can be observed in Table 3, incorporating external feedback from the environment or internal feedback from the world model consistently improves performance. Even though the world model does not provide as much improvement as the real environment, it also increases performance by over 3%. In particular, the synergy of both types of feedback yields the highest performance on most of the metrics, further confirming their effectiveness. In the following analysis, we will focus on our method using only environmental feedback for simplicity.

**Scaling of performance.** Here, we follow Brown et al. (2024); Snell et al. (2024); Wu et al. (2024b); OpenAI (2024) in considering the scaling w.r.t. inference computation. The results in Fig. 5a show that our method achieves better performance-computation tradeoff along with leading scaling w.r.t. inference computation. Thus, more inference budget can be allocated to improve its performance. Furthermore, in Fig. 5b, we show that its performance advantage is largely due to better long-horizon planning capabilities, achieving more than twice the success rate for extremely long plans.

**Computational efficiency.** Although our planner training is slower than baselines (36h vs. 12h) due to the equilibrium solving process for synthesizing training pairs (≈24h), it exhibits a competitive inference efficiency. For example, our method takes 16h to evaluate, while Tree-Planner takes 24h.

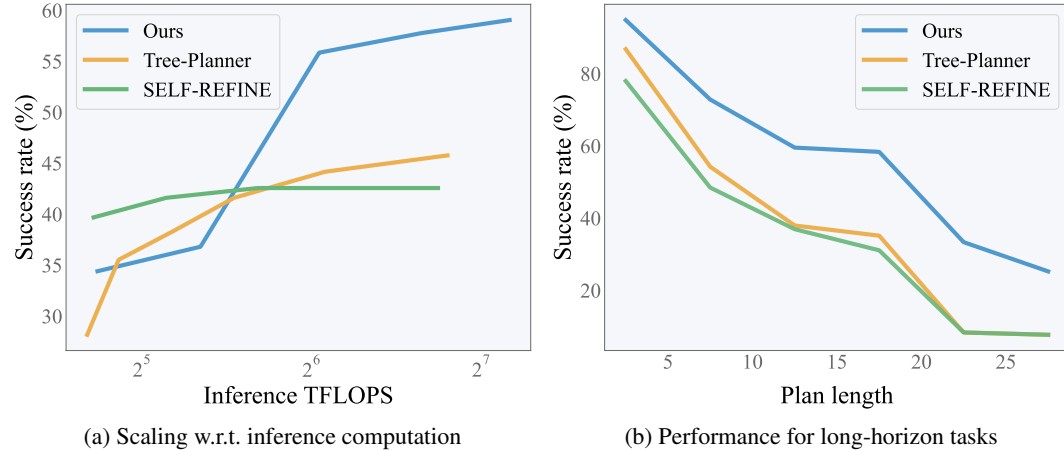

(a) Scaling w.r.t. inference computation

(b) Performance for long-horizon tasks

Figure 5: Performance analysis vs. inference computation and plan length. Our method shows leading scaling w.r.t. inference computation and long-horizon planning capabilities. The inference computation is measured by TFLOPS, and we consider KV cache when computing inference TFLOPS.

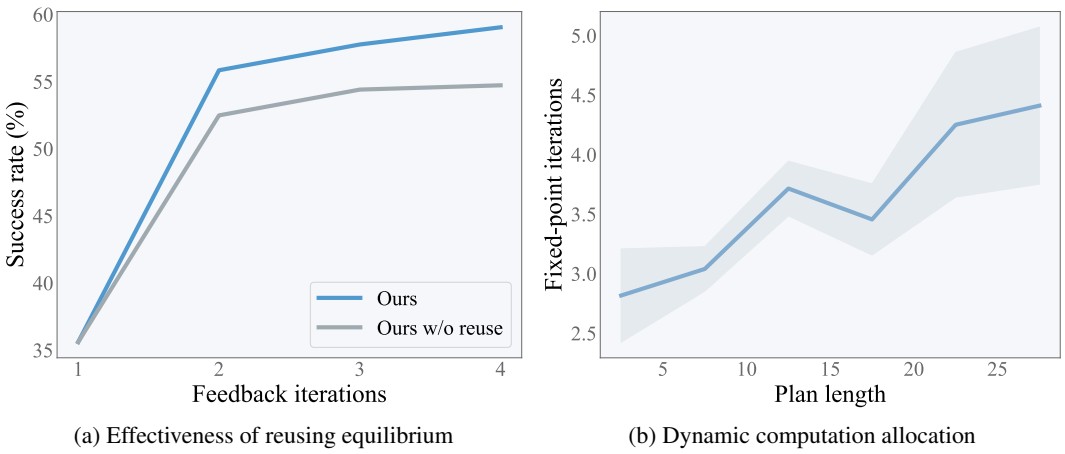

(a) Effectiveness of reusing equilibrium

(b) Dynamic computation allocation

Figure 6: Ablation study on computational efficiency. Our nested equilibrium solving process reuses previous equilibrium solutions and dynamically allocates the inference budget for better efficiency. The latter is displayed as the number of fixed-point iterations with mean and standard deviation.

This can be attributed to our design of reusing equilibrium in nested equilibrium solving, As illustrates in Fig. 6a, it accelerates the convergence of feedback iterations, achieving better performance ($>$55%) with significantly fewer interactions. Furthermore, we show in Fig. 6b that our method is able to dynamically allocate inference computation for planning tasks of different complexity.

## 5 CONCLUSION

This work proposes an equilibrium model-based LLM planner that is capable of self-refining plans from external and internal feedback. Unlike existing self-refinement methods based on prompting or sophisticated reinforcement learning, our proposed equilibrium sequence modeling allows simple supervised training of the self-refining planners. Moreover, it also enables the planner to efficiently incorporate environmental feedback or a world model for closed-loop planning. We implement the proposed approach on the VirtualHome-Env benchmark, and the experimental results suggest that it can dynamically allocate inference computation to achieve state-of-the-art planning performance.

**Reproducibility Statement.** Code is provided at `https://github.com/anonymous-iclr-2025/equilibrium-planner`. See Section 4.1 and Appendix B for the experimental settings.

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

# A BACKGROUND

## A.1 DEEP EQUILIBRIUM MODELS

Traditional neural networks are constructed by explicitly stacking layers $f^{(i)}$, which can be limited in their expressiveness due to the fixed number of layers and predetermined forward process. Instead, *implicit models* are defined by an underlying dynamic system to be solved, such as an ordinary differential equation (Chen et al., 2018), a controlled differential equation (Kidger et al., 2020), a stochastic differential equation (Kidger et al., 2021) or a fixed-point problem (Bai et al., 2019).

*Deep equilibrium models*, first introduced in Bai et al. (2019), are a representative class of implicit models characterized by fixed-point problems. Given an input $c$ and a function $f_\theta(x, c)$ such as a Transformer block (Bai et al., 2019) or a Transformer (Geng et al., 2023), deep equilibrium models define infinite-level stacking of this function $x_{i+1} = f_\theta(x_i, c)$ with $i = 0, 1, \ldots, L$ and $L \to \infty$ by solving the solution $x^*$ to the following fixed-point equation defined by $f_\theta$ and $c$:

$$x^* = f_\theta(x^*, c) \tag{11}$$

The forward pass of deep equilibrium models is root solving for the fixed-point problem. A common choice is the *fixed-point iteration* method, which starts from an initial guess $x_0$ and iteratively applies the transformation $x_{t+1} = f_\theta(x_t, c)$ until convergence. A sufficient condition for its convergence is if $f_\theta$ is a contraction mapping w.r.t. $x$, namely its Lipschitz constant is less than one (Banach, 1922), which could be relaxed by the well-posedness condition in El Ghaoui et al. (2021). More advanced root solvers include Broyden's method (Broyden, 1965) or Anderson acceleration (Anderson, 1965).

## A.2 TRAINING DEEP EQUILIBRIUM MODELS

Unlike traditional neural networks, whose gradient requires backpropagation through time (Werbos, 1990) at high memory and computational cost, the gradient of deep equilibrium models is computed analytically without differentiating over its forward pass. Given an equilibrium point $x^* = f_\theta(x^*, c)$ and a loss function $L(x^*, y)$, the loss gradient w.r.t. the model parameters $\theta$ is provided by the implicit function theorem (Krantz & Parks, 2002; Bai et al., 2019) as follows:

$$\frac{\partial L}{\partial \theta} = \frac{\partial L}{\partial x^*} \left( I - \frac{\partial f_\theta}{\partial x^*} \right)^{-1} \frac{\partial f_\theta}{\partial \theta}. \tag{12}$$

Its proof is given in Appendix A.4. Due to the challenge of exactly computing the inverse Jacobian term $A = (I - \frac{\partial f_\theta}{\partial x^*})^{-1}$ in the above gradient, existing work often approximate it via the damped fixed-point unrolling or the Neumann series (Geng et al., 2021b). Recently, Fung et al. (2022); Geng et al. (2021a) propose to approximate the inverse Jacobian term by $A \approx I$, with the former imposing strong assumptions. In practice, dropping the inverse Jacobian/Hessian has been used extensively in *one-step gradient* (Bolte et al., 2023; Luketina et al., 2016; Finn et al., 2017; Liu et al., 2019; Garima et al., 2020) and shown to be effective on Transformer-based LLMs (Choe et al., 2023).

## A.3 TRANSFORMER-BASED DEEP EQUILIBRIUM MODELS

Deep equilibrium models are initially proposed on Transformer architecture (Vaswani et al., 2017) for language modeling tasks (Bai et al., 2019). This seminal work considers a Transformer block as the basic unit $f_\theta$ in the equilibrium model. Then, Geng et al. (2021a) investigates improvements over the Transformer block by replacing self-attention with matrix decomposition. They also introduce one-step gradient based on the approximation of $A \approx I$ for efficiency and stability, assuming that the Lipschitz condition apply to a large number of matrix decomposition methods.

Recently, following the prevalence of Diffusion Transformers (Peebles & Xie, 2023), deep equilibrium models are extended to image generation tasks. Geng et al. (2023) propose generative equilibrium Transformers consisting of two modules, one using Transformer as the basic unit $f_\theta$ in the equilibrium model. Their method yields advanced one-step image generation results. (Bai & Melas-Kyriazi, 2024) replace most of the intermediate Transformer blocks with an equilibrium model, thus significantly reducing the number of parameters and memory usage for training and inference.

## A.4 PROOF OF IMPLICIT FUNCTION THEOREM

**Theorem 1** *(Implicit Function Theorem (Bai et al., 2019; Krantz & Parks, 2002)) Let $L : \mathbb{R}^n \times \mathbb{R}^n \to \mathbb{R}$ be a differentiable loss function, and let $f_\theta : \mathbb{R}^n \times \mathbb{R}^p \to \mathbb{R}^n$ be a differentiable function parameterized by $\theta \in \mathbb{R}^q$. Consider the following optimization problem:*

$$\min_\theta \quad L(x^*, y)$$
$$s.t. \quad x^* = f_\theta(x^*, c). \tag{13}$$

*where $x^*, y \in \mathbb{R}^n$, and $c \in \mathbb{R}^p$. If $\left(I - \frac{\partial f_\theta}{\partial x^*}\right)$ is invertible, then the loss gradient w.r.t. $\theta$ is given by:*

$$\frac{\partial L}{\partial \theta} = \frac{\partial L}{\partial x^*} \left(I - \frac{\partial f_\theta}{\partial x^*}\right)^{-1} \frac{\partial f_\theta}{\partial \theta}. \tag{14}$$

*Proof of Theorem 1.* To derive the loss gradient w.r.t. $\theta$, we begin by differentiating the equilibrium condition $x^* = f_\theta(x^*, c)$ with respect to $\theta$. Applying the chain rule, we have:

$$\frac{\partial x^*}{\partial \theta} = \frac{\partial f}{\partial \theta} + \frac{\partial f}{\partial x^*} \frac{\partial x^*}{\partial \theta}. \tag{15}$$

Given that $\left(I - \frac{\partial f_\theta}{\partial x^*}\right)$ is invertible, we can rearrange the above equation and solve for $\frac{\partial x^*}{\partial \theta}$:

$$\frac{\partial x^*}{\partial \theta} = \left(I - \frac{\partial f_\theta}{\partial x^*}\right)^{-1} \frac{\partial f_\theta}{\partial \theta}. \tag{16}$$

The chain rule implies $\frac{\partial L}{\partial \theta} = \frac{\partial L}{\partial x^*} \frac{\partial x^*}{\partial \theta}$. Substituting the expression for $\frac{\partial x^*}{\partial \theta}$, we obtain:

$$\frac{\partial L}{\partial \theta} = \frac{\partial L}{\partial x^*} \left(I - \frac{\partial f_\theta}{\partial x^*}\right)^{-1} \frac{\partial f_\theta}{\partial \theta}. \tag{17}$$

$\square$

## B  EXPERIMENTAL SETTINGS

### B.1  BENCHMARK

**Environment.** We adopt the robotic planning benchmark VirtualHome-Env (Liao et al., 2019) based on VirtualHome (Puig et al., 2018). It consists of a complex set of 292 planning tasks in 7 different indoor scenes, provided with 1360 mid-level action trajectories as ground truth annotations. These action trajectories are typically very long, with an average execution length of 10.8, highlighting its long-horizon characteristic. Moreover, the VirtualHome environment provides detailed feedback after performing each mid-level action, making it an ideal testbed for closed-loop planning.

Figure 7 visualizes a few examples sampled from the VirtualHome-Env benchmark. In each example, the planner is placed in an environment that spans a few indoor rooms and is given a detailed description of the environment. The description is originally in the form of a scene graph with objects as nodes and spatial relationships as edges, but for simplicity we present the planner with only the object nodes. The planner is then asked to generate a semantic action sequence based on a short task description. For instance, after receiving an instruction "turn on TV …", the robot agent must first walk to the table and grab the remote control, and then point at the TV to turn it on. As can be seen, these planning tasks usually involve a rather complex scene setup, and the ground truth action sequences are quite long. We provide more detailed statistics for this benchmark in Fig. 8.

Compared to alternative embodied planning benchmarks, VirtualHome-Env features long time horizons as well as more diverse closed-loop feedback. For example, ALFRED (Shridhar et al., 2020) and ReALFRED (Kim et al., 2024) are two common embodied instruction following benchmarks, but their plan lengths are relatively short and can be determined by a few templates, making them unsuitable for long-horizon planning. PlanBench (Valmeekam et al., 2023) and TravelPlanner (Xie et al., 2024) are recent benchmarks designed specifically for LLM planning, but they do not provide

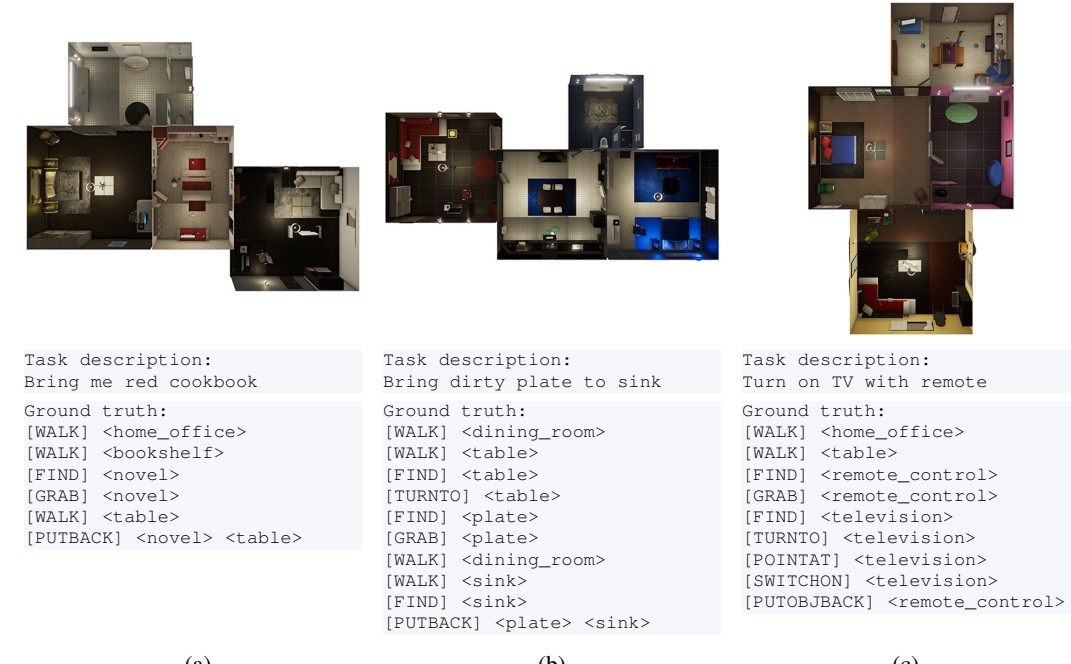

```
Task description:                Task description:                Task description:
Bring me red cookbook            Bring dirty plate to sink        Turn on TV with remote

Ground truth:                    Ground truth:                    Ground truth:
[WALK] <home_office>             [WALK] <dining_room>             [WALK] <home_office>
[WALK] <bookshelf>               [WALK] <table>                   [WALK] <table>
[FIND] <novel>                   [FIND] <table>                   [FIND] <remote_control>
[GRAB] <novel>                   [TURNTO] <table>                 [GRAB] <remote_control>
[WALK] <table>                   [FIND] <plate>                   [FIND] <television>
[PUTBACK] <novel> <table>        [GRAB] <plate>                   [TURNTO] <television>
                                 [WALK] <dining_room>             [POINTAT] <television>
                                 [WALK] <sink>                    [SWITCHON] <television>
                                 [FIND] <sink>                    [PUTOBJBACK] <remote_control>
                                 [PUTBACK] <plate> <sink>
```

|  (a)  |  (b)  |  (c)  |

Figure 7: Examples in VirtualHome-Env (Puig et al., 2018; Liao et al., 2019). The planner is given a detailed description of the environment (specifically, the objects within each rooms), a short task instruction, and is asked to output a sequence of mid-level actions associated with the correct objects.

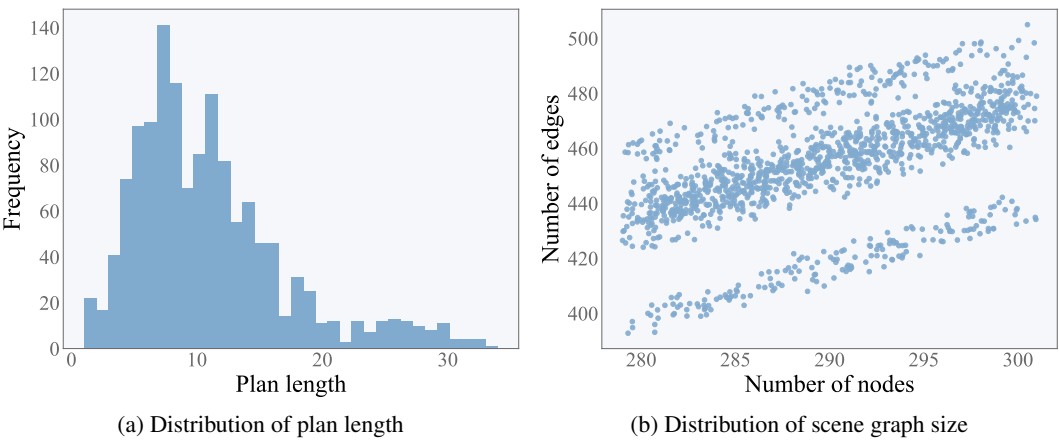

(a) Distribution of plan length          (b) Distribution of scene graph size

Figure 8: Detailed statistics of VirtualHome-Env (Puig et al., 2018; Liao et al., 2019). The benchmark features (a) a large set of long-horizon plans with an average length of 10.8, and (b) 7 complex scenes containing more than 280 objects and more than 400 valid relations. For the sake of clarity, we have excluded the CLOSE and FACING relations, which are redundant for most planning tasks.

closed-loop feedback during execution, which is an essential element of robotic planning. Therefore, we adopt VirtualHome-Env during the experiments, with further details presented below.

**Action.** The VirtualHome environment (Puig et al., 2018) originally supported animating 12 atomic actions based on the Unity simulator, with the followup work VirtualHome-Env (Liao et al., 2019) adding support for more actions using a graph simulator. It currently supports 40 atomic actions, in which 21 actions can be animated through Unity. Each action is defined by an action name and some object arguments, and is implemented by prewritten code executors. In our experiments, we use a full set of 40 actions included in the VirtualHome-Env dataset, summarized as follows:

1. Actions without object association: SLEEP, STANDUP, WAKEUP.

2. Actions associated with one object: WALK, FIND, GRAB, WASH, WIPE, PULL, PUSH, POUR, TURNTO, POINTAT, WATCH, TOUCH, OPEN, CLOSE, RUN, SIT, READ, PUTON, PUTOFF, DROP, LIE, SWITCHON, SWITCHOFF, DRINK, LOOKAT, TYPE, CUT, PUTOBJBACK, EAT, RINSE, PLUGIN, PLUGOUT, GREET, SCRUB, SQUEEZE.

3. Actions associated with two objects: PUTIN, PUTBACK.

**Feedback.** Because the environment includes a graph simulator of the scene graph, it can respond quickly to actions, *e.g.* changing object attributes, and provide the updated scene graph at each step. In our experiments, we curate several types of closed-loop feedback based on these scene graphs, simulating coarse feedback that may be received in real-world situations. Specifically, we consider the following four categories of environmental feedback associated with task failure:

1. Program format feedback:"Your output does not conform to the required format", meaning that the generated action sequence does not conform to the required format.

2. Invalid command feedback: "Your output has an invalid command: ...", meaning that the generated action sequence has an illegal command line.

3. Execution feedback: "Your output is executed incorrectly in the environment.", meaning that the generated action sequence cannot be executed in the environment.

4. Task completion feedback: "You have not completed this task. The following objects and corresponding states do not meet the goals: ... The following objects have wrong relative position: ...", meaning that the generated action sequence cannot complete the task, with more details about the task failure.

**Dataset Split.** We randomly divide the VirtualHome-Env dataset into training set and test set in a 50:50 ratio. To analyze the generalizability of our method, we mainly study the following three subsets of the test set: novel scene set, novel task set, and novel scene and task set. For instance, the novel scene set consists of seen planning tasks on unseen scenes. Overall, the dataset contains 735 training trajectories, 468 trajectories within the novel task set, 95 trajectories within the novel scene set, 62 trajectories within the novel scene and task set. Our models are first trained on the training set for a fixed number of epochs and then evaluated on the three test subsets above.

## B.2 BASELINES

Our method is mainly compared with Tree-Planner (Hu et al., 2024) and SELF-REFINE (Madaan et al., 2023), both reproduced using Llama 3 8B Instruct (Dubey et al., 2024) in line with ours. The former traverses an action tree that is built by repeated plan sampling, while the latter relies on self-refinement. To reproduce them, we perform supervised finetuning of Llama 3 on the training split of VirtualHome-Env for the same number of epochs as our method, and then follow their original procedures for inference. For instance, Tree-Planner is reproduced with both settings $N \in \{25, 50\}$ in action tree construction. The system prompts they use are similar to ours in Fig. 9.

We also report the results summarized by Hu et al. (2024) for reference. They additionally considered ProgPrompt (Singh et al., 2023), Zero-shot Planner (Huang et al., 2022) and two self-refinement planners, Local Replan (Raman et al., 2022; Guo et al., 2023) and Global Replan (Shinn et al., 2023). Since these baselines were implemented by calling the GPT-3.5 API instead of finetuning Llama 3, we report them in the novel scene and task track for a relatively fair comparison. It is worth noting that they adopted a smaller subset of actions and feedback, and differed in the curation of partial observations of the environment. Therefore, their results are for reference only.

There are several robotic planning baselines that we have not compared due to large environmental differences. For example, LLM-MCTS (Zhao et al., 2023) is a representative tree-search (Yao et al., 2023a) based planner. It followed Watch-and-help (Puig et al., 2021) to generate a dataset of simple embodied tasks (mostly object rearrangement tasks), while our work considers a more complex set of planning tasks. Alternative planners based on symbolic scene graph (Zhu et al., 2021; Rana et al., 2023), code (Liang et al., 2023; Sun et al., 2023), or PDDL (McDermott, 2000; Liu et al., 2023; Guan et al., 2023) are less flexible and difficult to implement in our environment.

```
You need to act as a task planner, who first draft an initial sub-task sequence and then refine it
    in the next few iterations.
When the the draft sub-task sequence is Null, you should output the initial sub-task sequence.
When the the draft sub-task sequence is not Null, You should refine it based on the the draft sub-
    task sequence.
If you have previously generated some action sequences and tried to execute them in the environment,
    their feedback will be provided to you for reference.
Each sub-task can be one of the following form: 1. [action_name]; 2. [action_name] <object name 1> (
    object id 1); 3. [action_name] <object name 1> (object id 1) <object name 2> (object id 2).
The (object id) is used to tell the simulator which object the action should act on.
The number of arguments depends on the action type.
For action type 1, the available actions are: SLEEP, STANDUP, WAKEUP
For action type 2, the available actions are: WALK, FIND, GRAB, WASH, WIPE, PULL, PUSH, POUR, TURNTO
    , POINTAT, WATCH, TOUCH, OPEN, CLOSE, RUN, SIT, READ, PUTON, PUTOFF, DROP, LIE, SWITCHON,
    SWITCHOFF, DRINK, LOOKAT, TYPE, CUT, PUTOBJBACK, EAT, RINSE, PLUGIN, PLUGOUT, GREET, SCRUB,
    SQUEEZE
For action type 3, the available actions are: PUTIN, PUTBACK
All action_name of the sub-tasks must be chosen from the above actions.
You should output the sub-task sequence in succinct form.
You must output END after you have output the entire sub-task sequence.
```
System Prompt

```
Task name:
Grab some juice

Instructions:
I go to the fridge, and grab some juice out of it. I then get a glass, and pour the juice into the
    glass.
```
Task

```
There are 4 rooms, and you are an embodied character with ID 198 in bedroom with ID 199.
The objects in each room is as follows:

Room name: home_office
Room ID: 1
Object ID and name in this room:
28 hanger
73 mat
......
Room name: dining_room
Room ID: 100
Object ID and name in this room:
116 ceiling
2005 food_food
......
```
Env

```
Feedbacks from past executions:
Action sequence:
[WALK] <dining_room> (100)
[WALK] <cupboard> (132)
[FIND] <cupboard> (132)
[OPEN] <cupboard> (132)
[FIND] <cup> (1000)
[GRAB] <cup> (1000)
[CLOSE] <cupboard> (132)
[WALK] <freezer> (141)
[OPEN] <freezer> (141)
[FIND] <juice> (1001)
[GRAB] <juice> (1001)
[POUR] <juice> (1001) <cup> (1000)
[PUTOBJBACK] <juice> (1001)
[CLOSE] <freezer> (141)
[END]
Feedback:
You have not completed this task.
The following objects have wrong relative position: (1000, cup) and (128, table).
```
Feedback $c_t$

```
The draft sub-task sequence:
[WALK] <dining_room> (100)
[WALK] <cupboard> (132)
[FIND] <cupboard> (132)
[OPEN] <cupboard> (132)
[FIND] <cup> (1000)
[GRAB] <cup> (1000)
[CLOSE] <cupboard> (132)
[WALK] <freezer> (141)
[OPEN] <freezer> (141)
[FIND] <juice> (1001)
[GRAB] <juice> (1001)
[POUR] <juice> (1001) <cup> (1000)
[PUTOBJBACK] <juice> (1001)
[CLOSE] <freezer> (141)
[END]
```
Draft Plan $x_t$

Figure 9: Example of the prompt used by our equilibrium planner.

## B.3 IMPLEMENTATION DETAILS

**Prompt.** Our approach involves two LLMs, one LLM as the equilibrium planner and an additional LLM as an optional world model. The planner's prompt is illustrated in Fig. 9. As can be seen, it consists of system prompt, task definition, environment description, history feedback, and draft plan.

Notably, the system prompt is modified from Hu et al. (2024), and the environment section describes the initial environment sorted by rooms, including the object names with their IDs within each room. For the additional world model, we adopt a similar prompt, except that it receives more information about the initial environment, including edges in the scene graph that correspond to spatial relations The world model is then asked to predict the environmental feedback given a generated plan.

**Finetuning.** Both our equilibrium planner and the world model are finetuned in a supervised manner. The equilibrium planner is finetuned for 6 iterations with a learning rate of 0.0002. At each iteration, we curate training data by pairing fixed points and ground truths, where the fixed points are sampled from an equilibrium memory of past iterations. To prevent overfitting to the history equilibrium, a decay ratio of 0.5 is used when sampling from the fixed points of previous iterations. Thereafter, we update the model parameters using gradient descent according to Eq. (6) for one epoch per iteration. For the world model, we collect all interacting experiences between the planner and the environment, including plans and feedback, and finetune it for 5 epochs using the same learning rate of 0.0002. Finetuning the world model takes about 30 hours due to its longer context (e.g. spatial relations).

**Inference.** The inference procedure of our planner was described in Algorithm 1, which involves a nested equilibrium solving process. Given the environment and the task instruction, we initialize the draft plan $x_0$ and the feedback $c_0$ as null and iterate through a nested loop. Each inner loop reuses the feedback from the outer loop to self-refine the draft plan, and after it converges, we update the feedback by interacting with the environment or world model. This process continues until it reaches an upper bound on the outer loop, which we set to 10 to match Tree-Planner (Hu et al., 2024). We adopt greedy sampling for the LLM, except that the first refinement step uses top-k sampling with $k = 10$ for higher diversity. Since most of the prompt remains unchanged during equilibrium solving, we implement KV cache to accelerate the inference process, which can be further improved with parallel decoding techniques (Santilli et al., 2023; Cai et al., 2024; Kou et al., 2024).

## C  ADDITIONAL RESULTS

To further illustrate the effectiveness of equilibrium sequence modeling in closed-loop planning, we exemplify the self-correction iterations on two long-horizon robotic planning task in Figs. 10 and 11. Compared to SELF-REFINE (Madaan et al., 2023) and Tree-Planner (Hu et al., 2024), our approach is more competent in revising a long plan through few forward passes without additional system 2. This is attributed to our efficient training scheme for teaching planners to self-refine.

We also compare different types of feedback utilized by our planner in Fig. 12. As can be observed, internal feedback alone cannot enable successful replanning, but it can reduce environmental interactions prior to convergence. This confirms the effectiveness of both internal and external feedback in closed-loop planning and thus justifies the framework design of our equilibrium planner. Furthermore, Fig. 13 shows the self-refining ability of our method at each inner-loop iteration. Even though the planner only succeeded in later steps, there is a clear quality improvement in its output during inner-loop introspection without additional feedback. Fig. 14 also validates this through a comparison with prompting-based self-refinement, where our model proves more advantageous.

## D  LIMITATIONS

While our equilibrium sequence modeling improves the planning capability of LLMs, we identify the following failure scenarios during the experiments: (1) hallucination of the equilibrium planner and the world model as in vanilla LLMs; (2) lack of awareness of history context such as previously grabbed objects. The latter may be resolved with reasoning techniques as in Yao et al. (2023b).

In a broader sense, our method may be limited in generalizing to new domains because it requires the ground truth and environmental feedback during training. These procedures with the equilibrium solving process results in lower training efficiency. Also, the current formulation only considers the explicit output plan without implicit reasoning steps. Furthermore, our model has only text input and no visual input, which limits its applicability in the real world. This can be resolved by introducing video-based planners (Du et al., 2024) and world models (Yang et al., 2024; Brooks et al., 2024).

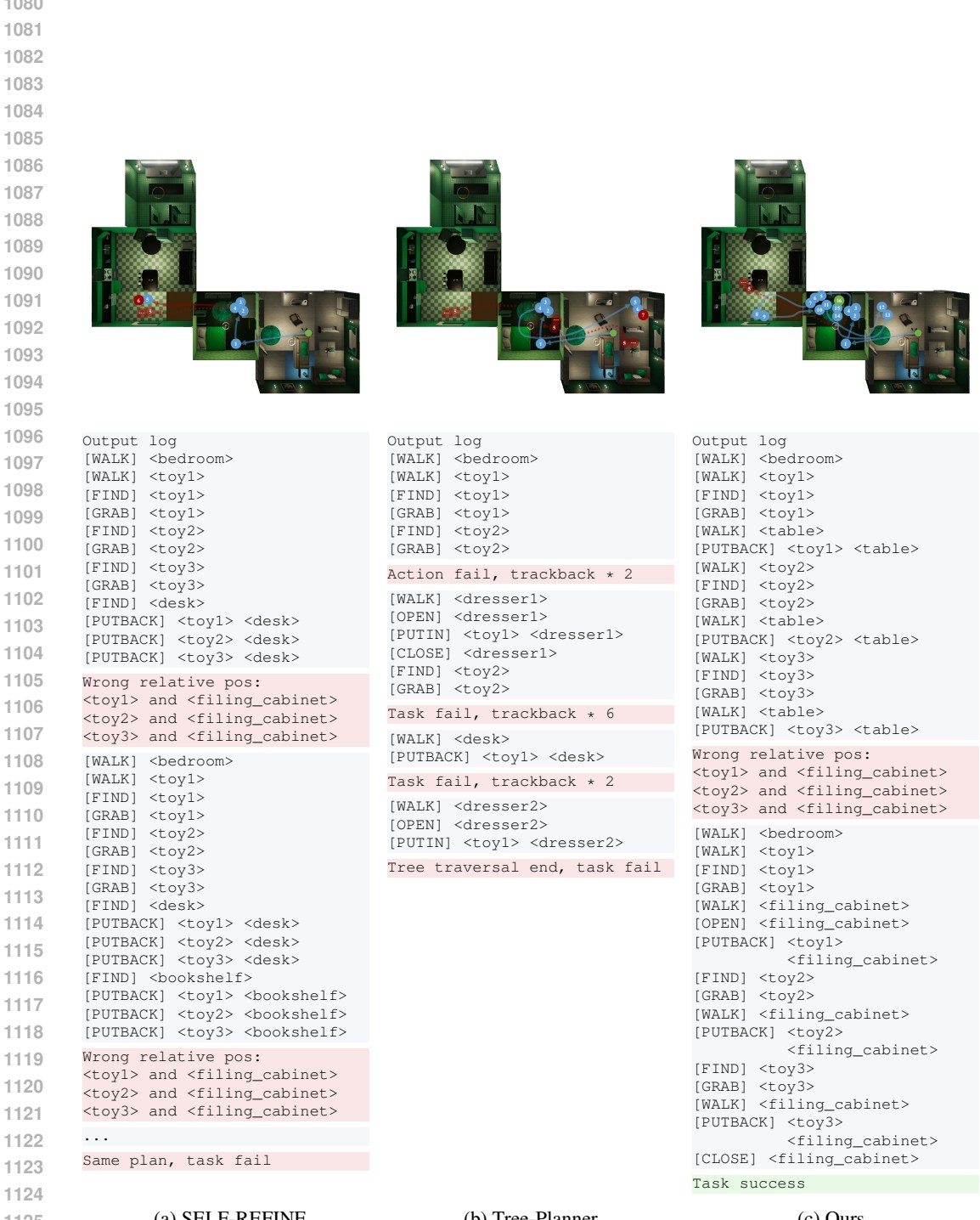

```
Output log                     Output log                     Output log
[WALK] <bedroom>               [WALK] <bedroom>               [WALK] <bedroom>
[WALK] <toy1>                  [WALK] <toy1>                  [WALK] <toy1>
[FIND] <toy1>                  [FIND] <toy1>                  [FIND] <toy1>
[GRAB] <toy1>                  [GRAB] <toy1>                  [GRAB] <toy1>
[FIND] <toy2>                  [FIND] <toy2>                  [WALK] <table>
[GRAB] <toy2>                  [GRAB] <toy2>                  [PUTBACK] <toy1> <table>
[FIND] <toy3>                                                 [WALK] <toy2>
[GRAB] <toy3>                  Action fail, trackback * 2      [FIND] <toy2>
[FIND] <desk>                                                 [GRAB] <toy2>
[PUTBACK] <toy1> <desk>        [WALK] <dresser1>              [WALK] <table>
[PUTBACK] <toy2> <desk>        [OPEN] <dresser1>             [PUTBACK] <toy2> <table>
[PUTBACK] <toy3> <desk>        [PUTIN] <toy1> <dresser1>     [WALK] <toy3>
                              [CLOSE] <dresser1>            [FIND] <toy3>
Wrong relative pos:            [FIND] <toy2>                 [GRAB] <toy3>
<toy1> and <filing_cabinet>    [GRAB] <toy2>                 [WALK] <table>
<toy2> and <filing_cabinet>                                  [PUTBACK] <toy3> <table>
<toy3> and <filing_cabinet>    Task fail, trackback * 6
                                                             Wrong relative pos:
[WALK] <bedroom>               [WALK] <desk>                 <toy1> and <filing_cabinet>
[WALK] <toy1>                  [PUTBACK] <toy1> <desk>       <toy2> and <filing_cabinet>
[FIND] <toy1>                                                <toy3> and <filing_cabinet>
[GRAB] <toy1>                  Task fail, trackback * 2
[FIND] <toy2>                                                [WALK] <bedroom>
[GRAB] <toy2>                  [WALK] <dresser2>             [WALK] <toy1>
[FIND] <toy3>                  [OPEN] <dresser2>             [FIND] <toy1>
[GRAB] <toy3>                  [PUTIN] <toy1> <dresser2>     [GRAB] <toy1>
[FIND] <desk>                                                [WALK] <filing_cabinet>
[PUTBACK] <toy1> <desk>        Tree traversal end, task fail [OPEN] <filing_cabinet>
[PUTBACK] <toy2> <desk>                                      [PUTBACK] <toy1>
[PUTBACK] <toy3> <desk>                                              <filing_cabinet>
[FIND] <bookshelf>                                           [FIND] <toy2>
[PUTBACK] <toy1> <bookshelf>                                 [GRAB] <toy2>
[PUTBACK] <toy2> <bookshelf>                                 [WALK] <filing_cabinet>
[PUTBACK] <toy3> <bookshelf>                                 [PUTBACK] <toy2>
                                                                    <filing_cabinet>
Wrong relative pos:                                          [FIND] <toy3>
<toy1> and <filing_cabinet>                                  [GRAB] <toy3>
<toy2> and <filing_cabinet>                                  [WALK] <filing_cabinet>
<toy3> and <filing_cabinet>                                  [PUTBACK] <toy3>
                                                                    <filing_cabinet>
...                                                          [CLOSE] <filing_cabinet>

Same plan, task fail                                         Task success
```

(a) SELF-REFINE        (b) Tree-Planner        (c) Ours

Figure 10: Visualization of our self-correction process in comparison with baselines. This example uses only environmental feedback, and we include toy IDs in the presentation for clarity. The task instruction is "Pick up all the toys on the floor and put them in their correct storage bin or shelf".

```
Output log
[WALK] <home_office>
[WALK] <laptop>
[FIND] <laptop>
[OPEN] <laptop>
[FIND] <mail>
[GRAB] <mail>
[FIND] <desk>
[OPEN] <desk>
[PUTBACK] <laptop> <desk>
[PUTBACK] <mail> <desk>
[FIND] <keyboard>
[TYPE] <keyboard>

Wrong state:
<character> {SITTING}
Wrong relative pos:
<character> and <mail>
<character> and <chair

...

Same plan, task fail
```

```
Output log
[WALK] <home_office>
[WALK] <laptop>
[FIND] <laptop>
[OPEN] <laptop>

Action fail, trackback * 1

[GRAB] <laptop>
[FIND] <chair>
[SIT] <chair>
[SWITCHON] <laptop>
[FIND] <mail>
[GRAB] <mail>
[READ] <mail>
[FIND] <keyboard>
[TYPE] <keyboard>

Task fail, trackback * 3

[FIND] <desk>
[OPEN] <desk>

Action fail, trackback * 7

[WALK] <table>
[OPEN] <laptop>

Action fail, trackback * 2

[WALK] <desk>
[OPEN] <laptop>

Action fail, trackback * 3

[TURNTO] <laptop>
[LOOKAT] <laptop>
[SWITCHON] <laptop>
[FIND] <chair>
[SIT] <chair>
[FIND] <mail>
[GRAB] <mail>
[READ] <mail>
[FIND] <keyboard>
[TYPE] <keyboard>

Task success
```

```
Output log
[WALK] <home_office>
[WALK] <laptop>
[FIND] <laptop>
[SWITCHON] <laptop>
[TURNTO] <laptop>
[LOOKAT] <laptop>
[FIND] <mail>
[READ] <mail>
[TURNTO] <mail>
[POINTAT] <mail>
[GRAB] <mail>
[FIND] <keyboard>
[TYPE] <keyboard>

Wrong state:
<character> {SITTING}
Wrong relative pos:
<character> and <mail>
<character> and <chair>

[WALK] <home_office>
[WALK] <laptop>
[FIND] <laptop>
[SWITCHON] <laptop>
[TURNTO] <laptop>
[LOOKAT] <laptop>
[FIND] <chair>
[SIT] <chair>
[FIND] <mail>
[GRAB] <mail>
[READ] <mail>
[TURNTO] <mail>
[POINTAT] <mail>
[WRITE] <mail>
[SWITCHOFF] <laptop>

Invalid command:
[WRITE] <mail>

[WALK] <home_office>
[WALK] <laptop>
[FIND] <laptop>
[SWITCHON] <laptop>
[TURNTO] <laptop>
[LOOKAT] <laptop>
[FIND] <chair>
[SIT] <chair>
[TURNTO] <laptop>
[POINTAT] <laptop>
[TURNTO] <laptop>
[LOOKAT] <laptop>
[FIND] <mail>
[GRAB] <mail>
[READ] <mail>
[TYPE] <laptop>

Task success
```

(a) SELF-REFINE              (b) Tree-Planner              (c) Ours

Figure 11: Visualization of our self-correction process in comparison with baselines. This example uses only environmental feedback. The task instruction is "Open email application, open new emails and respond accordingly". Our proposed method succeeds with fewer external feedback.

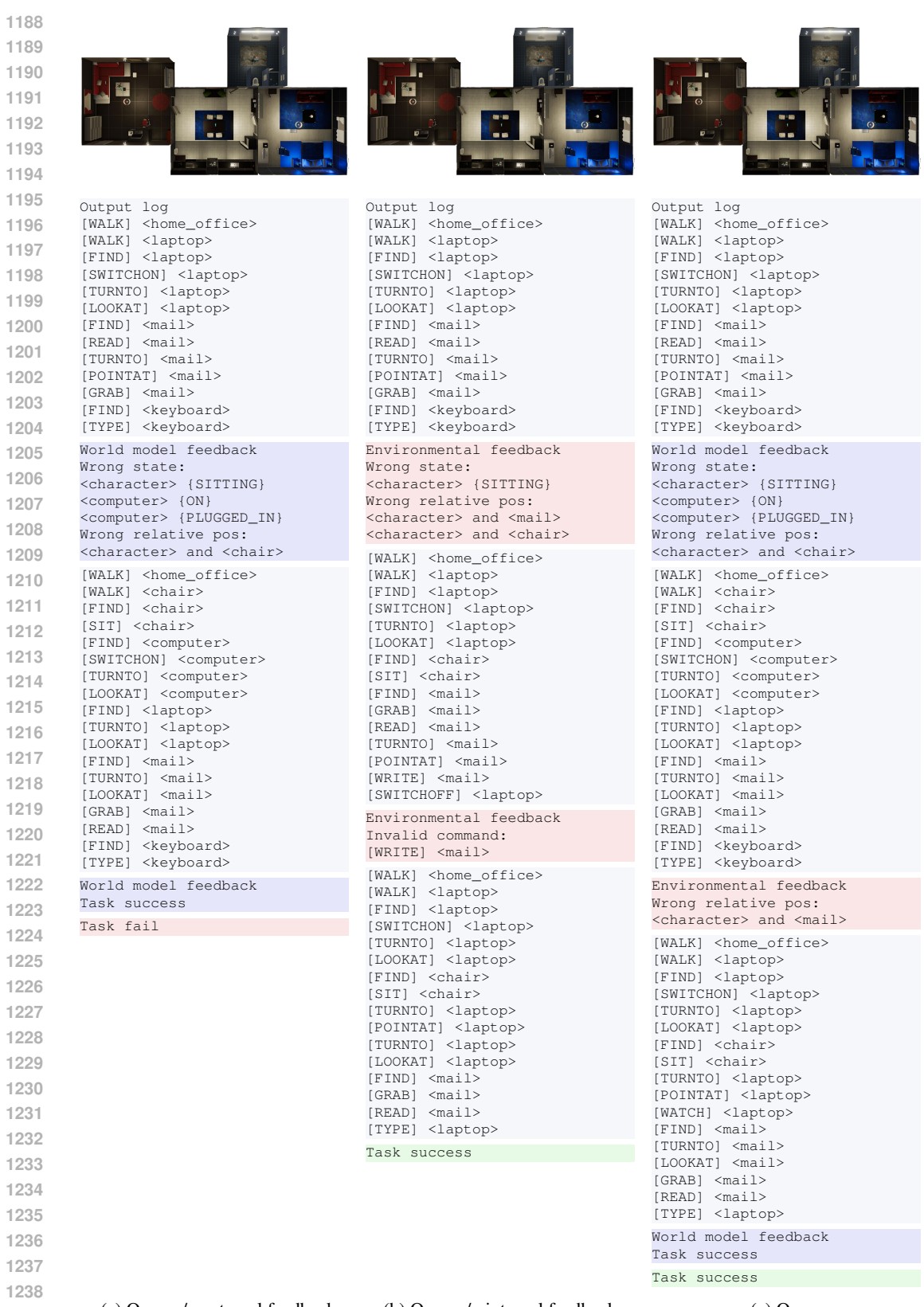

```
Output log
[WALK] <home_office>
[WALK] <laptop>
[FIND] <laptop>
[SWITCHON] <laptop>
[TURNTO] <laptop>
[LOOKAT] <laptop>
[FIND] <mail>
[READ] <mail>
[TURNTO] <mail>
[POINTAT] <mail>
[GRAB] <mail>
[FIND] <keyboard>
[TYPE] <keyboard>

World model feedback
Wrong state:
<character> {SITTING}
<computer> {ON}
<computer> {PLUGGED_IN}
Wrong relative pos:
<character> and <chair>

[WALK] <home_office>
[WALK] <chair>
[FIND] <chair>
[SIT] <chair>
[FIND] <computer>
[SWITCHON] <computer>
[TURNTO] <computer>
[LOOKAT] <computer>
[FIND] <laptop>
[TURNTO] <laptop>
[LOOKAT] <laptop>
[FIND] <mail>
[TURNTO] <mail>
[LOOKAT] <mail>
[GRAB] <mail>
[READ] <mail>
[FIND] <keyboard>
[TYPE] <keyboard>

World model feedback
Task success

Task fail
```

```
Output log
[WALK] <home_office>
[WALK] <laptop>
[FIND] <laptop>
[SWITCHON] <laptop>
[TURNTO] <laptop>
[LOOKAT] <laptop>
[FIND] <mail>
[READ] <mail>
[TURNTO] <mail>
[POINTAT] <mail>
[GRAB] <mail>
[FIND] <keyboard>
[TYPE] <keyboard>

Environmental feedback
Wrong state:
<character> {SITTING}
Wrong relative pos:
<character> and <mail>
<character> and <chair>

[WALK] <home_office>
[WALK] <laptop>
[FIND] <laptop>
[SWITCHON] <laptop>
[TURNTO] <laptop>
[LOOKAT] <laptop>
[FIND] <chair>
[SIT] <chair>
[FIND] <mail>
[GRAB] <mail>
[READ] <mail>
[TURNTO] <mail>
[POINTAT] <mail>
[WRITE] <mail>
[SWITCHOFF] <laptop>

Environmental feedback
Invalid command:
[WRITE] <mail>

[WALK] <home_office>
[WALK] <laptop>
[FIND] <laptop>
[SWITCHON] <laptop>
[TURNTO] <laptop>
[LOOKAT] <laptop>
[FIND] <chair>
[SIT] <chair>
[TURNTO] <laptop>
[POINTAT] <laptop>
[TURNTO] <laptop>
[LOOKAT] <laptop>
[FIND] <mail>
[GRAB] <mail>
[READ] <mail>
[TYPE] <laptop>

Task success
```

```
Output log
[WALK] <home_office>
[WALK] <laptop>
[FIND] <laptop>
[SWITCHON] <laptop>
[TURNTO] <laptop>
[LOOKAT] <laptop>
[FIND] <mail>
[READ] <mail>
[TURNTO] <mail>
[POINTAT] <mail>
[GRAB] <mail>
[FIND] <keyboard>
[TYPE] <keyboard>

World model feedback
Wrong state:
<character> {SITTING}
<computer> {ON}
<computer> {PLUGGED_IN}
Wrong relative pos:
<character> and <chair>

[WALK] <home_office>
[WALK] <chair>
[FIND] <chair>
[SIT] <chair>
[FIND] <computer>
[SWITCHON] <computer>
[TURNTO] <computer>
[LOOKAT] <computer>
[FIND] <laptop>
[TURNTO] <laptop>
[LOOKAT] <laptop>
[FIND] <mail>
[TURNTO] <mail>
[LOOKAT] <mail>
[GRAB] <mail>
[READ] <mail>
[FIND] <keyboard>
[TYPE] <keyboard>

Environmental feedback
Wrong relative pos:
<character> and <mail>

[WALK] <home_office>
[WALK] <laptop>
[FIND] <laptop>
[SWITCHON] <laptop>
[TURNTO] <laptop>
[LOOKAT] <laptop>
[FIND] <chair>
[SIT] <chair>
[TURNTO] <laptop>
[POINTAT] <laptop>
[WATCH] <laptop>
[FIND] <mail>
[TURNTO] <mail>
[LOOKAT] <mail>
[GRAB] <mail>
[READ] <mail>
[TYPE] <laptop>

World model feedback
Task success

Task success
```

(a) Ours w/o external feedback    (b) Ours w/o internal feedback    (c) Ours

Figure 12: Visualization of our self-correction process with different types of feedback. The external and internal feedback refer to the feedback from the environment and the world model, respectively. The task instruction is "Open email application, open new emails and respond accordingly".

```
Output log
[WALK] <bathroom>
[WALK] <cleaning_solution>
[FIND] <cleaning_solution>
[GRAB] <cleaning_solution>
[FIND] <rag>
[GRAB] <rag>
[POUR] <cleaning_solution>
       <rag>
[PUTOBJBACK]
       <cleaning_solution>
[FIND] <sink>
[TURNTO] <sink>
[LOOKAT] <sink>
[TOUCH] <sink>

Inner self-refinement

[WALK] <bathroom>
[WALK] <cleaning_solution>
[FIND] <cleaning_solution>
[GRAB] <cleaning_solution>
[FIND] <rag>
[POUR] <cleaning_solution>
       <rag>
[PUTOBJBACK]
       <cleaning_solution>
[FIND] <sink>
[TURNTO] <sink>
[LOOKAT] <sink>
[TOUCH] <sink>

Inner loop convergence

Environmental feedback
Wrong relative pos:
<cleaning_solution> and <sink>
```

(a) The 1st inner loop

```
Output log
[WALK] <bathroom>
[WALK] <cleaning_solution>
[FIND] <cleaning_solution>
[GRAB] <cleaning_solution>
[FIND] <sink>
[POUR] <cleaning_solution>
       <sink>
[TURNTO] <sink>
[LOOKAT] <sink>
[TOUCH] <sink>

Inner self-refinement

[WALK] <bathroom>
[WALK] <cleaning_solution>
[FIND] <cleaning_solution>
[GRAB] <cleaning_solution>
[FIND] <sink>
[POUR] <cleaning_solution>
       <sink>
[WAIT]
[FIND] <rag>
[GRAB] <rag>
[WIPE] <rag> <sink>
[PUTOBJBACK] <rag>
[SWITCHON] <faucet>
[RINSE] <sink>
[SWITCHOFF] <faucet>

Inner loop convergence

Environmental feedback
invalid command:
[WAIT]
```

(b) The 2nd inner loop

```
Output log
[WALK] <bathroom>
[WALK] <cleaning_solution>
[FIND] <cleaning_solution>
[GRAB] <cleaning_solution>
[FIND] <sink>
[POUR] <cleaning_solution>
       <sink>
[WIPE] <sink>
[PUTOBJBACK]
       <cleaning_solution>

Inner loop convergence

Task success
```

(c) The 3rd inner loop

Figure 13: Visualization of our self-correction process with inner-loop traces. This example uses only environmental feedback. The task instruction is "Spray some cleaner and wait a minute then wipe it out". As can be seen, each inner loop improves planning quality: the first removes a redundant [GRAB] action, while the second adds a crucial [WIPE] action and other interesting details.

```
Output log
[WALK] <bathroom>
[WALK] <electric_shaver>
[FIND] <electric_shaver>
[GRAB] <electric_shaver>
[FIND] <mirror>
[TURNTO] <mirror>
[LOOKAT] <mirror>
[FIND] <chair>
[SIT] <chair>
[SWITCHON] <electric_shaver>
[FIND] <beard>
[TURNTO] <beard>
[POINTAT] <beard>
[FIND] <beard>
[WIPE] <beard>
```

Inner loop convergence

invalid command:
[FIND] <beard>
Task fail

(a) Prompt-based self-refinement

```
Output log
[WALK] <bathroom>
[WALK] <electric_shaver>
[FIND] <electric_shaver>
[GRAB] <electric_shaver>
[FIND] <mirror>
[TURNTO] <mirror>
[LOOKAT] <mirror>
[SWITCHON] <electric_shaver>
[RUN] <mirror>
[TURNTO] <mirror>
[LOOKAT] <mirror>
[SWITCHOFF] <electric_shaver>
[PUTOBJBACK] <electric_shaver>
```

Inner self-refinement

```
[WALK] <bathroom>
[WALK] <electric_shaver>
[FIND] <electric_shaver>
[GRAB] <electric_shaver>
[FIND] <mirror>
[TURNTO] <mirror>
[LOOKAT] <mirror>
[SWITCHON] <electric_shaver>
[SWITCHOFF] <electric_shaver>
```

Inner loop convergence

Task success

(b) Ours

Figure 14: Visualization of our self-correction process in comparison with prompting-based method without any feedback. The task instruction is "Pick up razor and shave yourself". As can be seen, the prompting-based model converges without any self-correction, while our model achieves success..

