# OpenReview forum: "Closed-Loop Long-Horizon Robotic Planning via Equilibrium Sequence Modeling"
_ICLR.cc/2025/Conference — Submitted to ICLR 2025_

### Official Review · Reviewer_y2Lr · 2024-11-03

**Soundness:** 2
**Presentation:** 3
**Contribution:** 3
**Rating:** 6
**Confidence:** 3

**Summary:**

This work proposes a self-refining framework for robotic task planning using language model agents. By iteratively refining draft plans until equilibrium, this method enables efficient closed-loop, long-horizon planning. Utilizing a nested equilibrium process, the approach effectively integrates environmental feedback, enhancing task performance and inference efficiency. Evaluated on the VirtualHome-Env benchmark, it demonstrates increased scalability in robotic planning scenarios, supported by an equilibrium-based training method that allows end-to-end learning in a simple supervised manner.

**Strengths:**

- The introspection inspired approach provides a significant benefit over prompting based methods for similar planning tasks.
- The approach is intuitive and provides a unique perspective on solving planning problems via equilibrium methods.

**Weaknesses:**

While an inference study of computation is presented, it is unclear whether the training budgets of the competing approaches are fairly represented in the context of the finetuning costs of the world model for the proposed method.

**Questions:**

1. Are there any iteration-wise results available to show the subsequent effects of feedback over time compared to the other approaches?
2. How is the world model initialized using an LLM?
3. What is the frequency of switching between real experience and synthetic world-model generated experience?
4. Minor typo: L293: “May to easier to train”

---

> ### Author Response · Authors · 2024-11-20
> **Official Comment by Authors**
>
> We deeply appreciate the reviewer for providing valuable feedback. Below are our responses to the raised concerns.
>
> [W1] Training budget considering world model.
>
> * The training cost for the world model is about 30 hours, which is a non-negligible overhead. We have revised Section 4.3 and Appendix B.3 to reflect that. For a more fair comparison, we extend Table 1 to include baselines that also use a world model as feedback. The results are summarized below, where our approach remains advantageous and shows the most significant improvement with world model feedback, confirming its effectiveness in closed-loop robotic planning.
>
>   | Method            | World | model        |      | Both      | novel     |      | Novel     | scene     |      | Novel     | task      |
>   | ----------------- | ----- | ------------ | ---- | --------- | --------- | ---- | --------- | --------- | ---- | --------- | --------- |
>   |                   |       |              |      | SR        | GCR       |      | SR        | GCR       |      | SR        | GCR       |
>   | Tree-Planner N=25 |       | x            |      | 38.71     | 63.18     |      | 51.58     | 69.45     |      | 40.38     | **63.75** |
>   | Tree-Planner N=25 |       | $\checkmark$ |      | 38.71     | 58.71     |      | 51.58     | 63.94     |      | 38.46     | 57.40     |
>   | Tree-Planner N=50 |       | x            |      | 38.71     | 63.50     |      | 51.58     | 69.54     |      | 39.74     | 63.29     |
>   | Tree-Planner N=50 |       | $\checkmark$ |      | 37.10     | 57.53     |      | 54.74     | 69.64     |      | 39.32     | 58.84     |
>   | Ours              |       | x            |      | 33.87     | 59.98     |      | 49.47     | 66.60     |      | 34.62     | 59.06     |
>   | Ours              |       | $\checkmark$ |      | **40.32** | **65.40** |      | **65.26** | **79.47** |      | **41.88** | 62.76     |
>
> [Q1] Subsequent effects of feedback.
>
> * Following your suggestion, we present iteration-wise results in Figures 11 and 13 in the appendix. As shown in Figure 11, our method improves stably over outer-loop iterations and succeeds with only two feedbacks. In comparison, Tree-Planner can ignore previous feedback and make the same mistake twice ([OPEN] \<laptop\>) because its tree-backtracking mechanism does not reuse the implicit knowledge in earlier feedback.
> * Figure 13 illustrates the continuing effect of feedback on our inner-loop iterations, where each iteration refines the plan while adhering to previous feedback. For example, the second inner loop inserts a critical [WIPE] action while maintaining all objects' relative positions according to the last feedback, leading to later success.
>
> [Q2] Initialization of world model.
>
> * The world model is initialized from Llama 3 8B Instruct and supervised finetuned on all the planner's environmental interactions. This procedure takes place after the equilibrium planner has completed training, so that all of its data can be leveraged at once. The finetuning data is constructed in a format similar to Figure 9 in the appendix, with a different order to predict feedback from the plan. See Appendix B.3 for more details.
>
> [Q3] Switch between real and generated experience.
>
> * Our planner alternates between real experience and generated experience at each outer-loop iteration. This process is intuitively illustrated in Figure 12c in the appendix. We note that it is possible to use generated experience more frequently and reduce the number of environmental interactions if the planner has access to a more accurate world model.
>
> [Q4] Minor typo.
>
> * Thank you for pointing this out, we have corrected the typo and will continue to revise the manuscript.

---

> > ### Comment · Reviewer_y2Lr · 2024-11-27
> >
> > I appreciate the clarifications provided by the authors and the responses to my questions. I maintain my assessment of the paper.

---

### Official Review · Reviewer_ZYWJ · 2024-11-04

**Soundness:** 2
**Presentation:** 1
**Contribution:** 2
**Rating:** 3
**Confidence:** 2

**Summary:**

This paper introduces a novel approach to long-horizon robotic task planning by addressing limitations in current large language model (LLM) planning agents, which often struggle with foresight and error correction. The authors propose an equilibrium sequence modeling technique that iteratively refines a plan until it reaches a stable state, or equilibrium, without requiring additional reward models or verifiers. This approach allows for efficient closed-loop planning by incorporating environmental feedback, enhancing the robot's ability to plan and adapt in complex tasks. Through self-refinement, the method improves long-horizon action prediction and reduces computation by reusing equilibrium states, resulting in superior performance on the VirtualHome-Env benchmark.

The model integrates a nested feedback system, which combines both internal (world model) and external (environmental) feedback to guide plan refinement, achieving improved executability and success rates compared to existing models. Unlike reinforcement learning-based methods, this equilibrium-based framework is trained in a supervised manner, simplifying training while enabling high computational efficiency. The results demonstrate that the proposed approach outperforms baseline methods on various metrics, indicating its potential for more robust and adaptive robotic task planning in dynamic environments.

**Strengths:**

If I have understood correctly, the core problem tackled by the paper is really crucial. A lot of approaches are using LLMs as high-level planners. However, multiple evaluations have shown that LLMs are unable to reason over long horizon. With the proposed approach of self introspection, LLMs can be used to aide the planning while *verifying* the LLM generated plan until the equilibrium is reached.

**Weaknesses:**

I do not have many specific weaknesses of the paper. The major reason for that (which is the biggest weakness and reason for my overall rating) is that the paper is extremely  hard to follow. I was not able to understand the paper properly even after reading it multiple times. As I mentioned in the summary of the paper, I was able to get the overall idea but the details of the approach are still unknown to me. However, here following are some ways in which the paper can be made readable:

- The paper is not self sufficient. It heavily relies on previous works about equilibrium models and fixed-point iteration methods. Readers who are not familiar with these previous works would fine nearly impossible to read the paper. The paper can be updated by having Background section that discusses these methods.
-  I did not understand how the self refinement with a fixed $c$ works? If the there is no feedback from the environment, how does the approach know even if the generated plan is correct or not in the first place.
-  There seems to be some distinction between the "training" and "equilibrium solving via fixed point iteration" but they also seem to be connected. It is not clear from the paper how are they different and how are they connected.
- How this is exactly connected with LLMs is not clear. What part of the modules are using LLMs are not clear from the paper. Can authors comment on it and make it clear in the paper?
- How does nested refinement work if the context is similar for the inner loop without any feedback? How does the equilibrium model know what to change between two plans?
- What are the inputs? What are the assumptions? This needs to be made clear in the paper.

In general, IMO the writing of the paper needs to be self-sufficient. Which the paper is currently not. Above I have tried to outline some of the points that can be used to update the readability of the paper.

**Questions:**

Please refer to the weaknesses section.

---

> ### Author Response · Authors · 2024-11-20
> **Official Comment by Authors**
>
> We deeply appreciate the time and effort you dedicated to reviewing our paper and providing constructive feedback. Below we address the raised concerns one by one.
>
> [W1] Background on equilibrium models and fixed-point iteration.
>
> * Following your suggestion, we have substantially expanded the **background introduction with Appendix A**, detailing the previous literature on deep equilibrium models and fixed-point iteration. We will continue our efforts to make the draft self-sufficient for readers who are unfamiliar with this line of research.
>
> [W2] Self-refinement with fixed context.
>
> * Even with fixed context, self-refinement is still beneficial in two ways: (1) Vanilla LLM planners are limited by unidirectional dependency, i.e. the previous tokens cannot attend to future tokens, resulting in limited ability to plan ahead. Self-refinement addresses this by allowing each output token to attend to the entire plan (from an older version). (2) Vanilla LLM planners use a fixed forward process with less flexible computation, while self-refinement enables dynamic allocation of inference compute through an iterative process.
>
> * To validate the effectiveness of our method for self-correction with fixed context, it is compared to a self-refinement baseline via prompting. As shown in the table below, our method clearly improves the success rate even without any feedback. We also illustrate the inference traces in Figure 14 in the appendix, where our method is shown to be more effective at steering the output toward a correct plan.
>
>   | Method | Both | novel | | Novel | scene | | Novel | task  |
>   | -- | -- | -- | -- | - | - | - | - | - |
>   | | SR | GCR | | SR | GCR | | SR | GCR |
>   | Supervised | 24.19 | 32.55 | | 41.05 | 49.81 | | 26.07 | 35.53 |
>   | SELF-REFINE | 32.26 | 52.29 | | 44.21 | 61.25 | | 30.98 | 51.80 |
>   | Ours | **33.87** | **59.98** | | **49.47** | **66.60** | | **34.62** | **59.06** |
>
> [W3] Distinction between training and equilibrium solving.
>
> * Our approach consists of two alternating phases: (1) solving for the equilibrium solutions, and (2) training with these equilibrium solutions. The two phases differ in that the first phase involves only forward passes defined by $x_{t+1}=f_\theta(x_t,c)$, while the second phase involves backpropagation of the loss $L(f_\theta(x^*,c),y)$. On the other hand, they are closely connected because the second phase relies on the equilibrium $x^*$ produced by the first phase.
>
> [W4] Connection with LLMs.
>
> * The LLMs correspond to $f_\theta$ in our equilibrium algorithm. Take an example, the equilibrium solving process actually performs an iterative inference of LLM based on the previous output tokens $x_t$ until the new output tokens $x_{t+1}=f_\theta(x_t,c)$ have converged. Following your suggestion, we have revised the main paper to explicitly declare $f_\theta$ as an LLM in Section 3.1.
>
> [W5] Self-refinement within inner loops.
>
> * We illustrate the effect of inner loops in self-refinement with Figure 13 in the appendix. Even without additional feedback, it shows consistent quality improvements during each inner loop. In particular, the first inner loop removes a redundant [GRAB] action, while the second loop adds a [WIPE] action that is crucial for later success, as well as interesting details such as [WAIT] and [RINSE] \<sink\>. Thus, inner loops are considered useful in our nested refinement framework.
> * The working mechanism of inner loops is the same as our previous answer to W2, i.e., introducing bidirectional dependency and using more computation. We also note its similarity to human introspection, both involving little external feedback but a lot of random attempts to improve in new directions (as shown in Figure 13).
>
> [W6] Inputs and assumptions.
>
> * The inputs to our LLM planner are described in Appendix B.3 and Figure 9. It consists of five parts, including a system prompt that asks the LLM to self-refine plans, a task description, an environment description, the latest draft plan, and all previous feedback. Please refer to Appendix B.3 and Figure 9 for more details.
> * The assumptions in our proposed method are twofold: (1) LLMs could converge to equilibrium by fixed-point iteration. This has been verified in Figure 6b, which shows that only a small number of iterations (between 2 and 5) is required for convergence. (2) For the implicit function theorem (Theorem 1) to hold, $(I-\frac{\partial f_\theta}{\partial x^*})$ should be invertible, i.e., full rank. This is commonly assumed in deep equilibrium models, including the work [1, 2] similarly based on Transformers. We have noted this assumption in Theorem 1.
>
> Overall, we have made several updates to the draft based on the reviewers' constructive feedback on self-sufficiency, and will continue to revise the draft to reflect all suggestions.
>
> ---
>
> References:
>
> [1] Bai, et al. Deep Equilibrium Models. NeurIPS 2019.
>
> [2] Geng, et al. One-Step Diffusion Distillation via Deep Equilibrium Models. NeurIPS 2023.

---

### Official Review · Reviewer_djD9 · 2024-11-05

**Soundness:** 1
**Presentation:** 3
**Contribution:** 2
**Rating:** 3
**Confidence:** 4

**Summary:**

The paper proposes a new LLM planning framework via equilibrium models. The goal is to build a long-horizon task planner that can incorporate feedback from the environment as well. The authors start with the limitations of prior LLM planners, such as the lack of bidirectional dependency, and lack of internal error correction. Built on these insights, the authors propose to make LLM planners iterative like equilibrium models. The direct training objective of the equilibrium model is intractable, so the authors proposed a simplified objective built on prior works in theory. The simplified objective involves running the LLM iteratively to a fixed point and imposing supervised training loss only on the fixed point. An associated planning framework is then proposed using the fine-tuned LLM. The authors present empirical results on VirtualHome and compared against previous LM planning frameworks.

**Strengths:**

The paper is well written. The proposed concept is relatively simple and straightforward. The math is easy to follow. The intuition of self-refinement is grounded.

**Weaknesses:**

My strongest criticism of of paper is the core math it builds on: the authors say one can simplify Eqn(4) to Eqn (5) per Theorem 3.1 in Fung et al. (2022) under appropriate assumption. Yet the "appropriate assumption" is completely unsatisfied here which makes the entire math perspective unfounded. Specifically, Theorem 3.1 in Fung et al. (2022) on their Assumptions 2.1, 3.1, 3.2, and 3.3; Let's take Assumption 2.1 for example, the mapping used in equilibrium model ($T_\theta$ in Eqn (6) of Fung et al. (2022) and $f_\theta$ in the reviewing manuscript), must be Lipschitz with respect to input. This is absolutely ungrounded for transformer-based LLMs. In fact, ML theory papers always devote a great amount of effort to circumvent this problem just because of how unacceptable it is to assume this transformer. This is not to mention assumptions 3.1, 3.2, and 3.3 in Fung et al. (2022). Therefore, you are not even making a strong assumption, but an assumption that's widely considered wrong for LLMs.

Therefore, I urge all reviewers to review the paper by ignoring the flawed math explanation and examining its merit from its empirical merit.

The technical contribution becomes very limited if we proceed to ignore flaws in the training objective. The feedback, iterative planner seems to be an obvious solution even considering the environmental feedback perspective.

I think one straightforward way for the authors to address the flawed training objective is to conduct ablations that uses your planning framework but with original Llama3 8B that's not fine-tuned. This is no way making the math correct but can inform us about the performance of your method without a unfounded training objective.

I also wonder whether the training procedures in line 204 and 205 are grounded. Specifically, how practical it is to run an LM to reach a fixed point? It seems to me that in a high-dim space like language token sequences, it's really hard to achieve convergence. Is it because the authors specifically prompted it to do so?

**Questions:**

How practical it is to run an LM to reach a fixed point? How sensitive is convergence with respect to the initial sequence?

How would your method perform without fine-tuning Llama model?

The authors suggested, "they divided the dataset into a training set and three test subsets, including the novel scene set, the novel task set, and the novel scene and task set." Since VirtualHome is a popular benchmark, have you surveyed prior papers about their partitioning of the tasks? Have you attempted using a more conventional partitioning used by prior works?

---

> ### Author Response · Authors · 2024-11-20
> **Official Comment by Authors**
>
> We sincerely appreciate the time and effort you dedicated to reviewing our paper and providing valuable feedback. Below are our responses to the raised concerns.
>
> [W1] Justification for approximating $A=(I-\frac{\partial f_\theta}{\partial x^*})^{-1}\approx I$.
>
> * This approximation does not rely on Theorem 3.1 of Fung et al. (2022). Dropping the inverse Jacobian is a common practice in one-step gradient (see Appendix A.2), and has been shown to be effective on Transformer-based LLMs [1]. We thank the reviewer for pointing out the incorrect reference; indeed, it is difficult to get theoretical guarantees for large Transformers. We have revised the draft to remove that reference and to include more relevant evidence [1].
>
> * **This approximation does not affect our core contribution** in teaching self-refinement via supervised training. In fact, the original Equation (4) before this approximation can already transform self-refinement into a supervised learning problem $\min_{\theta}L(A^\top f_\theta(x^*,c)+(I-A^\top)x^*,y)$ of its equilibrium, bypassing any process supervision or reinforcement learning. This work adopts approximation $A\approx I$ in the trade-off between accuracy and efficiency, opting for the latter. We expect further improvements from more accurate approximation techniques given sufficient computational budgets.
>
> [W2] Limited technical contribution.
>
> * The technical contributions of this paper are twofold: (1) for LLM planning research, it proposes a new supervised training method for self-fining LLM planners; (2) for deep equilibrium model research, it proposes a series of novel designs (nested equilibrium solving, reuse of feedback, etc.) to improve the training and inference efficiency of deep equilibrium models.
>
> [W3/Q2] Ablation of training objective
>
> * To validate the effectiveness of our new training objective, we compare our finetuned LLM with two baselines (the original Llama 3 8B Instruct using our prompts, and self-refinement based on a vanilla supervised fintuned Llama 3 8B) with the setup of Table 2. The results below show that the effectiveness of our method is largely due to the new training objective, while the original or supervised finetuned Llama does not work well.
>
>   | Method | Both | novel |  | Novel | scene | | Novel | task |
>   | - | - | - | - | - | - | - | - | - |
>   | | SR        | GCR       |      | SR        | GCR       |      | SR        | GCR       |
>   | Original Llama + our prompt | 0.00      | 0.00      |      | 1.05      | 1.05      |      | 1.07      | 1.07      |
>   | SELF-REFINE                 | 43.55     | 65.18     |      | 54.74     | 70.24     |      | 39.96     | 62.37     |
>   | Ours                        | **56.45** | **76.63** |      | **58.95** | **87.07** |      | **54.91** | **74.18** |
>
> [W4/Q1] Fixed-point convergence.
>
> * The following table shows the number of iterations for an LLM to reach its fixed point, aggregated over 10 initial plan for each of 60 random tasks. All LLMs considered, including the original Llama, the supervised finetuned Llama, and our model, can converge to a fixed point within a few iterations, regardless of the initial sequence. This is partly due to our greedy sampling strategy (described in Appendix B.3) that reduces the LLMs' randomness, after which they tend to repeat themselves (Figures 4a, 10a, and 11a) and converge easily.
>
>   | Method | Avg  | stat | | |
>   | - | - | - | - | - |
>   | | Mean | Std  | Min  | Max   |
>   | Original Llama | 6.70 | 2.12 | 3.22 | 14.98 |
>   | SFT Llama      | 2.46 | 0.88 | 2.02 | 3.80  |
>   | Ours           | 3.02 | 1.61 | 2.32 | 4.07  |
>
> [Q3] Dataset partitioning.
>
> * We have surveyed prior papers using VirtualHome in Appendix B.2. There are two recent dataset protocols represented by LLM-MCTS [2] and Tree-Planner [3]. Specifically, LLM-MCTS uses the indoor scenes of VirtualHome and synthesizes its own dataset focusing on object rearrangement tasks. Tree-Planner uses a subset of VirtualHome-Env to evaluate for training-free methods. Since both works have not released their detailed dataset, we simply use the original VirtualHome-Env dataset and partition it accordingly in our experiments.
>
>   | Method | Public | #Tasks | #Scenes | Task content  | Task splits |
>   | - | - | - | - | - | - |
>   | Tree-Planner | x  | 35     | 4       | household     | Novel scene and task                                         |
>   | LLM-MCTS     | x            | 2000   | 4       | rearrangement | Novel scene and task, Novel scene, Novel task, Seen scene and task |
>   | Ours         | $\checkmark$ | 1360   | 7       | household     | Novel scene and task, Novel scene, Novel task                |
>
> ---
>
> References:
>
> [1] Choe, et al. Making Scalable Meta Learning Practical. NeurIPS 2023.
>
> [2] Zhao, et al. Large Language Models as Commonsense Knowledge for Large-Scale Task Planning. NeurIPS 2023.
>
> [3] Hu, et al. Tree-Planner: Efficient Close-loop Task Planning with Large Language Models. ICLR 2024.

---

### Official Review · Reviewer_jqoA · 2024-11-07

**Soundness:** 3
**Presentation:** 4
**Contribution:** 3
**Rating:** 6
**Confidence:** 3

**Summary:**

This paper proposes formulating long-horizon task planning with LLMs as a fixed point problem, arguing that such a formulation leads to a natural realization of many desirable properties of task planning, including global context when deciding the plan, the ability to recover from errors in drafts of the plan, and the ability to scale inference time compute to yield better plans. The authors draw from the literature of deep equilibrium models [1] to instantiate their idea, proposing a training framework based off of the implicit differentiation algorithm proposed by [1]. The full framework proposed by the authors includes a nested inference algorithm, with an inner loop that solves the fixed point problem for a static context, and an outer loop that sends the plan to the environment for feedback. In the case that the environment cannot be queried online, the authors propose training a world model to simulate environment feedback. In comparisons with baselines, the proposed approach performs the best on the VirtualHome-Env benchmark.

[1] Bai, Shaojie, J. Zico Kolter, and Vladlen Koltun. "Deep equilibrium models." Advances in neural information processing systems 32 (2019).

**Strengths:**

1. As the authors mention in their introduction, there is indeed a natural match between the problems faced by task planning (and more generally long-horizon reasoning) and the benefits posed by deep equilibrium models. There appears to be substantial originality in the authors' identification of this match, and their subsequent application of deep equilibrium models to long-horizon task planning.
2. The experimental evaluations are fairly thorough, with many ablation studies I wanted to see present.
3. The manuscript is clear and well written.

**Weaknesses:**

1. The first row in table 3 seems to correspond to the setting in which no correction is allowed and the synthetic world model is not used (so only the inner loop is run). The results obtained by the proposed algorithm in this setting are however lower than the Tree-Planner baseline results from table 1. What explains this low performance? More generally, why is the synthetic world model needed, and could baseline approaches benefit as well from such a world model?
2. Although prior work has solely evaluated on the VirtualHome-Env benchmark, the results would be more solidified if the authors had chosen an additional benchmark to evaluate on.

**Questions:**

1. The authors cite [1] as evidence that self-refinement via prompting tends to not lead to successful error corrections. Figure 4 shows an example of error correction by the proposed method with environment feedback (i.e., with both the inner and outer loop). Does self-refinement via fixed point iteration (i.e., only running the inner loop, making it more similar to the setting of [1]) lead to better error correction performance than the approaches studied in [1]? If so, then it would be nice to see some empirical evidence of this or a few sample inference traces.
2. For the experimental results presented in table 2, what is the ratio of the number of calls to the synthetic world model to the number of environment interactions?
3. How does the dynamic allocation of inference compute presented in figure 6b work? What decides how many inference forward passes to take?
4. Is the dataset used for training constructed once, or adaptively generated from the newest solutions to the equilibrium problem?
5. Why can the proposed approach improve better from environment feedback than baselines?

[1] Jie Huang, Xinyun Chen, Swaroop Mishra, Huaixiu Steven Zheng, Adams Wei Yu, Xinying Song, and Denny Zhou. Large language models cannot self-correct reasoning yet. In International Conference on Learning Representations, 2024.

---

> ### Author Response · Authors · 2024-11-20
> **Official Comment by Authors**
>
> Thank you for recognizing our work and providing valuable feedback. Below are our responses to the raised concerns.
>
> [W1.1/Q1] Self-refinement without external feedback or world model.
>
> * In the setting without feedback, our performance is inferior to Tree-Planner due to the different number of plans generated. Tree-Planner always generates 25 or 50 candidate plans at a time before constructing the action tree, while our method refines over a single plan with a few iterations (<10). This gives Tree-Planner a comparative advantage. However, in the other settings that allow feedback (e.g. Tables 2), our method outperforms Tree-Planner by incorporating feedback more flexibly.
>
> * To validate the effectiveness of our method for self-correction without feedback, it is compared to a self-refinement baseline via prompting. As shown in the table below, our method clearly improves the success rate even without any feedback. We also illustrate the inference traces in Figure 14 in the appendix, where our method is shown to be more effective at steering the output toward a correct plan.
>
>   | Method      | Both      | novel     |      | Novel     | scene     |      | Novel     | task      |
>   | ----------- | --------- | --------- | ---- | --------- | --------- | ---- | --------- | --------- |
>   |             | SR        | GCR       |      | SR        | GCR       |      | SR        | GCR       |
>   | Supervised  | 24.19     | 32.55     |      | 41.05     | 49.81     |      | 26.07     | 35.53     |
>   | SELF-REFINE | 32.26     | 52.29     |      | 44.21     | 61.25     |      | 30.98     | 51.80     |
>   | Ours        | **33.87** | **59.98** |      | **49.47** | **66.60** |      | **34.62** | **59.06** |
>
> [W1.2] Usage of world model.
>
> * We need the world model as an efficient alternative to the environment. Since environmental interactions can be expensive and sometimes unrecoverable, the world model helps the planner self-correct with fewer environmental interactions (Figure 12), thereby improving the performance under a low interaction budget. This is also confirmed in Table 3 of the main paper, where the world model is shown to be beneficial in both of two experimental settings.
>
> * For a more fair comparison, we extend Table 1 to include baselines that also use a world model as feedback. The results are summarized below, where our approach remains advantageous and shows the most significant improvement with world model feedback, confirming its effectiveness in closed-loop robotic planning.
>
>   | Method            | World | model        |      | Both      | novel     |      | Novel     | scene     |      | Novel     | task      |
>   | ----------------- | ----- | ------------ | ---- | --------- | --------- | ---- | --------- | --------- | ---- | --------- | --------- |
>   |                   |       |              |      | SR        | GCR       |      | SR        | GCR       |      | SR        | GCR       |
>   | Tree-Planner N=25 |       | x            |      | 38.71     | 63.18     |      | 51.58     | 69.45     |      | 40.38     | **63.75** |
>   | Tree-Planner N=25 |       | $\checkmark$ |      | 38.71     | 58.71     |      | 51.58     | 63.94     |      | 38.46     | 57.40     |
>   | Tree-Planner N=50 |       | x            |      | 38.71     | 63.50     |      | 51.58     | 69.54     |      | 39.74     | 63.29     |
>   | Tree-Planner N=50 |       | $\checkmark$ |      | 37.10     | 57.53     |      | 54.74     | 69.64     |      | 39.32     | 58.84     |
>   | Ours              |       | x            |      | 33.87     | 59.98     |      | 49.47     | 66.60     |      | 34.62     | 59.06     |
>   | Ours              |       | $\checkmark$ |      | **40.32** | **65.40** |      | **65.26** | **79.47** |      | **41.88** | 62.76     |
>
> [W2] Additional benchmark.
>
> * We have briefly surveyed other planning benchmarks (ALFRED, PlanBench, etc.) in the Environment section of Appendix B.1. As shown, VirtualHome remains a more suitable benchmark for closed-loop long-horizon planning, since the other benchmarks focus on either closed-loop or long-horizon aspects. Therefore, this paper only validates the proposed planner on VirtualHome. Nevertheless, we believe that closed-loop long-horizon planning may be implicitly involved in broader applications such as web agents and video generation, and we will continue to explore this direction in future work.

---

> ### Author Response · Authors · 2024-11-20
> **Official Comment by Authors (Continued)**
>
> [Q2] Ratio of environmental interactions to world model calls.
>
> * Their ratio is currently set to 1:1, i.e. the planner alternates between using the environmental feedback and the internal feedback from the world model at each outer-loop iteration. It is possible to reduce this ratio and the number of environmental interactions required if the planner has access to a more accurate world model.
>
> [Q3] Dynamic allocation of inference compute
>
> * Our method dynamically allocates the inference budget by performing iterative inference until the output converges. This process is automatic, and we only intervene when the number of iterations reaches an upper bound of 10 (which is rarely reached). Thus, the number of forward passes is mostly determined by the convergence speed of the model itself. Interestingly, Figure 6b shows that this number correlates with the length of the target plan, indicating that the model is able to dynamically allocate more inference compute for more complex plans.
>
> [Q4] Construction of training data.
>
> * The training data is constructed adaptively using the newest equilibrium solutions. Specifically, an equilibrium memory is maintained that buffers all equilibrium solutions, including the newest ones. At each iteration, we curate the training data by weighted sampling from this memory (where the newest solutions are sampled more frequently) and then pairing them with the ground truths. See the Finetuning section in Appendix B.3 for more details.
>
> [Q5] Rationale behind better improvement from feedback.
>
> * Compared to tree-based alternatives, self-refinement is more flexible. Self-refinement takes into account feedback through forward passes of LLMs, allowing arbitrary changes based on the LLMs' knowledge. This approach also enables the model to correct multiple errors in parallel (Figure 10c). In contrast, tree-based alternatives require backtracking in a tree, which is costly when correcting an early mistake and does not fully exploit the verbalized knowledge in feedback.
> * Compared to self-refinement via prompting, our training-based approach is more effective. This is because our training objective $L(f_\theta(x^*,c),y)$ directly teaches the LLM to self-refine by mapping a suboptimal equilibrium plan $x^*$ and the feedback $c$ to a better plan $y$. By associating plan refinement with past feedback during training, the LLM learns to take feedback into account and self-refine in a more meaningful direction.

---

> ### Comment · Reviewer_jqoA · 2024-11-26
> **Response to Rebuttal**
>
> I thank the authors for their response. My score remains the same.

---

### Meta-Review · Area_Chair_Et4P · 2024-12-20

**Metareview:**

The paper proposes an approach to closed-loop planning for long-horizon robotic tasks using equilibrium sequence modeling and LLM-based self-refinement. While the empirical results are compelling, the reviewers identified several significant concerns.

Strengths:

- The paper tackles the critical problem of long-horizon planning with LLMs.

- Innovative use of equilibrium models for plan refinement.

Weaknesses:

- Reviewers share valid concerns about flawed mathematical justifications. The reliance on unsupported assumptions diminishes the strength of the theoretical contribution.

- Some reviewers noted that the paper is challenging to follow for those unfamiliar with equilibrium models. Improved explanations and a self-contained presentation are necessary.

- Questions remain about the training cost, dataset partitioning, and the mechanisms for integrating feedback.

The paper shows great promise but requires substantial revisions to address theoretical concerns, improve clarity, and solidify its contributions. I would recommend rejection in its current form.

**Additional Comments On Reviewer Discussion:**

The paper received mixed reviews during the initial round. Some reviewers highlighted significant concerns about the flawed mathematical justification of the proposed method, the paper's lack of clarity and accessibility, and unresolved questions about implementation details, such as training budgets and feedback mechanisms. In response, the authors revised the paper by removing incorrect references, expanding the background sections, and providing additional experimental results. Despite these efforts, the fundamental issues of theoretical soundness and insufficient clarity remained unresolved, as noted in the updated reviews. Ultimately, the reviewers reached a consensus, with those who initially gave higher ratings lowering their scores.

---

### Decision · Program_Chairs · 2025-01-22

Reject